# Reviews and synthesis on increasing hypoxia in eastern boundary upwelling systems: zooplankton under metabolic stress

Leissing Frederick[1,3], Mauricio A. Urbina[2, 3] and Ruben Escribano[3,4]

[1] Doctoral Program in Oceanography, University of Concepcion, Chile
[2] Department of Zoology, Faculty of Natural Sciences and Oceanography, University of Concepcion, Chile
[3] Millennium Institute of Oceanography, University of Concepcion, Chile
[4] Department of Oceanography, Faculty of Natural Sciences and Oceanography, University of Concepcion, Chile

*Correspondence to*: Ruben Escribano (rescribano@udec.cl)

**Abstract.** Eastern boundary upwelling systems (EBUS) are ecologically and economically important marine regions of the world ocean. In these systems, zooplankton play a pivotal role in transferring primary production up through the food web. Recent studies show that global warming is causing a gradual deoxygenation of the world ocean, while in EBUS an expansion

and intensification of the subsurface oxygen minimum zone (OMZ) is taking place further exacerbating hypoxic conditions for zooplankton inhabiting the upwelling zone. Hypoxia can affect zooplankton by limiting their aerobic respiration, and constraining migration, energy budget, reproduction, and development. These effects, however, depend on some specific adaptations evolved in habitats, permanently or episodically, subjected to low oxygen waters. Various metabolic, physiological, behavioural, and morphological adaptations have been described in zooplankton interacting with the OMZ.

Adjustment on the aerobic respiration under variable oxygen levels deserves special attention, since such adaptive responses to endure mild or severe hypoxia may involve trade-offs in energy usage that impact other metabolic functions or energy-demanding processes. In addition, the oxidative stress resulting from exposure to highly fluctuating oxygen conditions in the upwelling zone can impose further energy expenses.  New demands imply a reduction in the energy budget otherwise available for, escape, migration, growth, feeding and reproduction with further ecological consequences for population and community

dynamics. This paper reviews and explores the existence or lack of such adaptive metabolic responses along with potential effects of oxidative stress, and their role for zooplankton dynamics in EBUS with major consequences for the pelagic food web and biological productivity.

## 1 INTRODUCTION

It is widely recognized that the increase in atmospheric $CO_2$ and other greenhouse gasses is driving the warming of the Earth's surface and ocean (Oschlies et al., 2018) with several other physical consequences, such as increased events of strong winds, storms in some regions, and changes in ocean circulation (Schmidtko et al., 2017). The warming of the upper layers of the ocean drives greater stratification of the water column, reduces vertical mixing, and affects ocean ventilation. Additionally, a warmer ocean lowers oxygen solubility and increase metabolic demands of ectotherms (Breitburg et al., 2018), further challenging marine life. In this scenario, deoxygenation has led to approximately a 2% loss of oxygen in the open ocean and coastal waters since the mid-20th century (Stramma et al., 2008; Schmidtko et al., 2017; Breitburg et al., 2018).

The persistent and expanding oxygen minimum zones (OMZs), characterized by extremely low oxygen concentrations (<20-45 µmol kg⁻¹) (Stramma et al., 2008), and associated with highly productive coastal and oceanic regions (Gilly et al., 2013), are becoming increasingly critical as they reach shallower depths. This shoaling of the upper boundary, along with the descent of the lower boundary, has led to a vertical expansion and increased total volume of OMZs (Stramma et al., 2010). These changes reduce the vertical habitat available for zooplankton, limiting their distribution and potentially altering their ecological roles (Köhn et al., 2022). Additionally, further reductions in oxygen concentrations within OMZ cores have been observed, intensifying their effects on aerobic organisms (Chan et al., 2008). Also, extreme events caused by mesoscale eddies can produce intense episodes of hypoxia in the upwelling zone (Köhn et al., 2022) and even transport OMZ waters off the coast (Auger et al., 2021).

In some areas, mainly at mid latitudes, of the four major eastern boundary current systems (EBUS) (Chavez and Messié, 2009), the effect of climate change has been associated with an intensification of the physical forcings driving coastal upwelling (Bakun et al., 2010; Xiu et al. 2018; Bograd et al., 2023), leading to several changes in the physical-chemical properties of the water column, including a gradual cooling in the last few decades (Santos et al., 2012; Schneider et al., 2016). However, other studies have found no evidence of increasing upwelling or trends in upwelling intensity, based on time series observations for several decades in same EBUS (e.g. Pardo et al., 2011; Bode et al., 2019). Trends of upwelling intensity in EBUS is therefore still matter of controversy, and the predictive models reveal much uncertainty on the future of upwelling regarding its spatial and temporal variability (Bograd et al., 2023). Upon a potential increase of upwelling, alongshore winds in EBUS bring colder water and more frequent occurrences of upwelling events (Breitburg et al., 2018), although some modelling work also suggests an extension of the upwelling period and spatial homogenization of upwelling alongshore (Wang et al., 2015). Stronger upwelling is ultimately thought to be a response to the strengthening of large-scale pressure gradients linked to global-scale climate change (Garcia-Reyes and Largier, 2009). The closely linked effects of potentially increasing upwelling, cooling of the water column and shoaling of the OMZ in EBUS driven by global warming are illustrated in Fig. 1.

The ongoing combined processes, deoxygenation, increasing upwelling, and OMZ expansion will alter the oxygen conditions in upper layers (<50 m) in EBUS, where plankton concentrates, having various ecological and biogeochemical consequences. In this respect, Ekau et al. (2010) demonstrated that hypoxic conditions can alter the zooplankton community composition in

the Benguela EBUS. This due to variable tolerances to hypoxia in distinctive groups, being euphausiids, for example, better adapted to low oxygen (<0.1 mL L$^{-1}$) compared to copepods. Escribano et al. (2009) also described a strong vertical zonation of zooplankton depending on variable tolerance to hypoxia in the northern upwelling zone of Chile. Variable tolerance to hypoxia is also reflected in some species-dependent physiological rates of copepods, as found in the calanoids *A. tonsa* and *C. chilensis* whose egg production rate and hatching success were strongly and positively correlated to oxygen concentration under laboratory conditions (Ruz et al., 2015; Choi et al., 2024). In the same context, not only vertical distribution, but also the vertical amplitude of the diel vertical migration can be strongly modulated by hypoxic conditions forced by the position of the OMZ core and its upper boundary (Tutasi and Escribano, 2020; Riquelme-Bugueño et al., 2020; Wishner et al., 2020). Aerobic metazooplankton inhabiting the upwelling zone is thus expected to be exposed to variable levels of oxygenation from normoxia to mild or sever hypoxia, depending on their distribution and migrating behaviour. Their responses will also depend on the existence, absence, or development of new adaptations. In this paper, we review such adaptive responses of zooplankton and the ecological consequences driven by hypoxia, aiming at establishing the physiological/metabolic bases and directions when addressing issues related to the future of zooplankton dynamics in EBUS subjected to ongoing climate change.

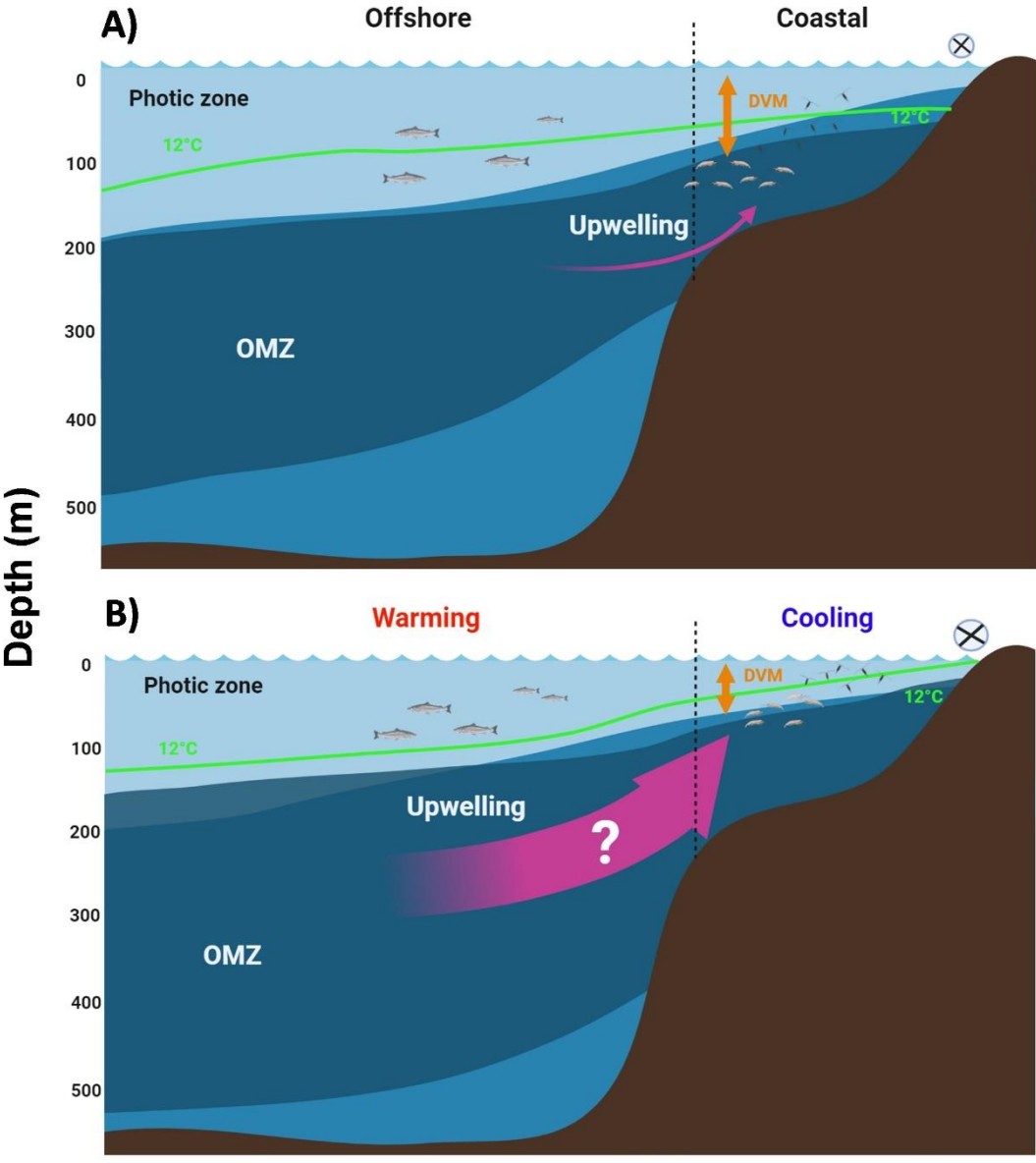

**Figure 1. Projected effects of the expansion of the OMZ in eastern boundary upwelling systems (EBUS).** A) Under present (initial) conditions, wind-driven upwelling raises the OMZ system and brings cold-water into shallow depths at the inshore as illustrated by the 12°C isotherm and so fertilizes the photic zone and promotes plankton aggregation. B) Ocean warming effects manifest mainly at surface in the offshore region, while a vertically expanded OMZ along with an eventual (although uncertain) increase in upwelling may cool down the coastal zone and further shoaling the OMZ at the inshore area,

spreading hypoxia and vertically reducing the oxygenated habitat. Vertical compression of the oxygenated habitat may restrict the vertical distribution and diel vertical migration (DVM) of zooplankton.

## 2 ADAPTIVE RESPONSES OF ZOOPLANKTON TO HYPOXIA

Oxygen plays a key role in the structuring and functioning of marine ecosystems and so modulates the spatial-temporal distribution of many marine organisms. This is mainly because low oxygen levels challenge the maintenance of aerobic metabolism and therefore challenging the survival for most of the biota (Vaquer-Sunyer and Duarte, 2008; Ekau et al., 2010; Wishner et al. 2018; Breitburg et al., 2018). The effects of depleted oxygen can affect organisms in many ways, including acute natatory and physiological impairment, diminished growth, and reproductive success, altered behaviour of mobile forms when searching for more favourable oxygen regimes (Wishner et al., 2018), reduced metabolic rates including respiration and ammonium excretion (Kiko et al., 2016), and lower recruitment due to reduction in egg hatching (Ruz et al., 2018; Choi et al., 2024).

The adaptation of animals to low oxygen has driven a strong selective pressure for maintaining aerobic metabolism, by optimizing and enhancing oxygen uptake from hypoxic waters (Childress and Seibel, 1998), or alternatively by suppressing their metabolic rate to reduce the oxygen demands (Seibel, 2011). At the upper end of the oxygen cascade, oxygen uptake is satisfied by two adaptive modes: as an oxygen-conformer organism by reducing aerobic metabolic rate as environmental oxygen decreases, or as an oxygen-regulator by maintaining the aerobic metabolism down to an oxygen level known as critical oxygen tension ($P_{crit}$)(Rogers et al., 2016). The difference between these two adaptive modes can be illustrated by the changes in the metabolic rate as a function of oxygen pressure (Fig. 2).

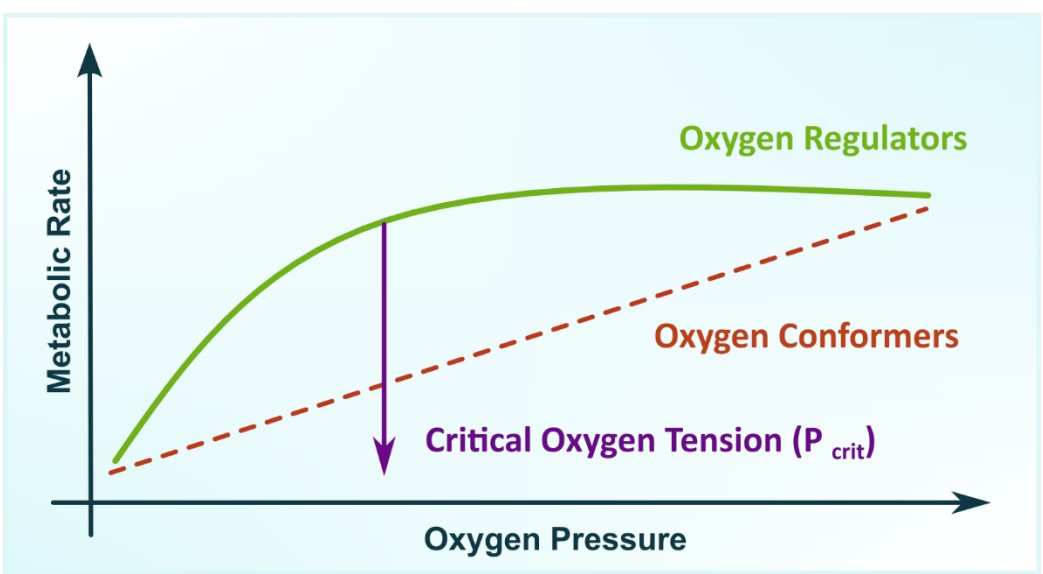

**Figure 2. Metabolic rate as a function of oxygen pressure**, illustrating two adaptive modes in marine zooplankton where an oxygen conformer organism reduces the aerobic metabolic rate as environmental oxygen decreases, while an oxygen regulator maintains aerobic metabolism down to an oxygen pressure known as critical oxygen tension ($P_{crit}$) (modified from Pörtner et al., 1993).

The adaptive responses to variable levels of oxygen illustrated in Fig. 2 may take place over short timescales (hours), as driven by the exposure to vertical gradients of oxygen in the water column when performing DVM in migrating species, or in the case of non-migratory species due to vertical mixing in the water column forced by upwelling pulses, or by changes in wind conditions promoting mixing, both processes over a synoptic (2-5 days) time scale in the upwelling zone (Sobarzo et al., 2007). Both migratory and non-migratory species can thus be exposed to variable oxygenation in the upwelling zone, which is characterized by a marked oxygen-stratified water column. This condition is well illustrated by the oxygen distribution in the water column of central-southern Chile throughout an annual cycle (Fig. 3). Within the photic zone (about 50 m in the coastal zone), the annual cycle of oxygen conditions reveals the existence of normoxic, mild and severely hypoxic habitats, which vertically migratory and non-migratory zooplankton must face depending on their vertical distribution (Fig. 3). In addition, shallower strata are more variable on dissolved oxygen during intense upwelling (above 30 m deep), while the deeper strata (30-50 m deep) are more variable in oxygen during a depressed upwelling in the winter.

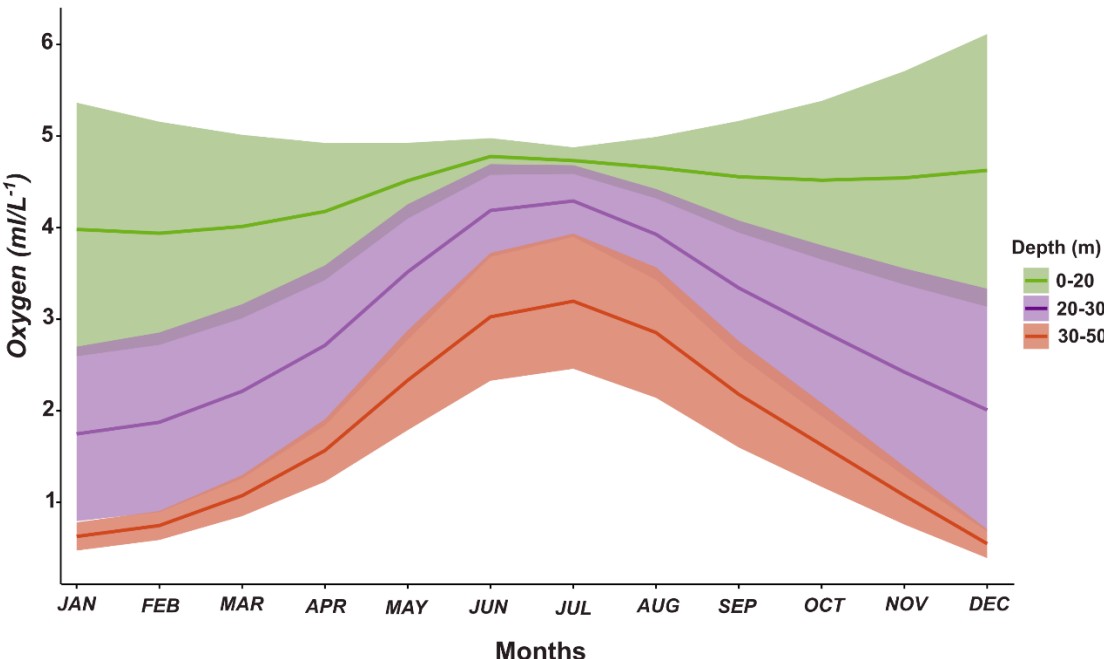

**Figure 3. The annual variability (monthly climatology) of dissolved oxygen in three strata in the upper 50 m layer off central-southern Chile (36°30'S).** Oxygen data are from the time series study at Station 18 off central-southern Chile during the period 2002-2016 (Frederick et al., 2024). Shallower strata exhibit greater variability during intense upwelling in summer, while deeper strata show more variability during depressed upwelling in winter. The shaded area corresponds to the overlap of the distribution range of the organisms.

Furthermore, the response to oxygen variation illustrated in Fig. 2, may promote different metabolic adaptations, for instance by reducing $P_{crit}$ to cope with severe hypoxia, or a lack of adjusting capacity. This adaptive response has been shown to vary among copepod species (Frederick et al., 2024) and can therefore favour some species above others that are negatively impacted. Such differential species-specific responses are shown in Fig. 4 which describes the metabolic rate as a function of oxygen condition from normoxia to hypoxia. A non-adaptive response (Fig. 4A) is reflected in a direct relationship between the metabolic rate (MR) and oxygen levels, which also implies a change in $P_{crit}$ (fixed α). In contrast, Adaptive Response 1 (Fig. 4B) represents a constant MR at the cost of changing $P_{crit}$ (variable α), while Adaptive Response 2 (Fig. 4C) maintains a constant $P_{crit}$ with a consequent change in MR (also variable α) as a function of oxygen levels. Among these, only Adaptive Response 1 (Fig. 4B) does not compromise aerobic scope, allowing for better regulation of physiological processes throughout the year (Fig. 3), but at expenses of not taking advantage of favourable periods such as phytoplankton blooms. In contrast,

organisms with Adaptive Response 2 can better capitalize on favourable conditions in a changing ocean (likely spring and summer blooms) at the expense of reducing their aerobic scope.

It has recently been suggested that $P_{crit}$ is the point at which the oxygen supply capacity ($\alpha$) reaches its maximum physiological limit (uptake and delivery) and as such is a species- and temperature-specific value (Siebel et al., 2021). For instance: $P_{crit}$ is constant within a given fish species (Rogers et al., 2016) and thus supporting the validity of the maximum oxygen supply capacity within species. Yet, the potential plasticity of $P_{crit}$ has rarely been explored on invertebrate inhabiting fluctuating oxygen regimes, such a copepods living around OMZ zones (Wishner et al., 2018; Frederick et al., 2024). Furthermore, many of these pelagic invertebrates, which can produce several cohorts throughout the year, combine maternal effects with plasticity, so that promoting shifts in either $P_{crit}$ or MR in some species of planktonic copepods depending on season during the year cycle (Frederick et al., 2024). Such differential responses will ultimately alter the species composition with consequences for the food web structure. However, even for those oxygen-regulator species, severe hypoxia may be lower than $P_{crit}$ resulting in severe stress or deleterious effects on organisms.

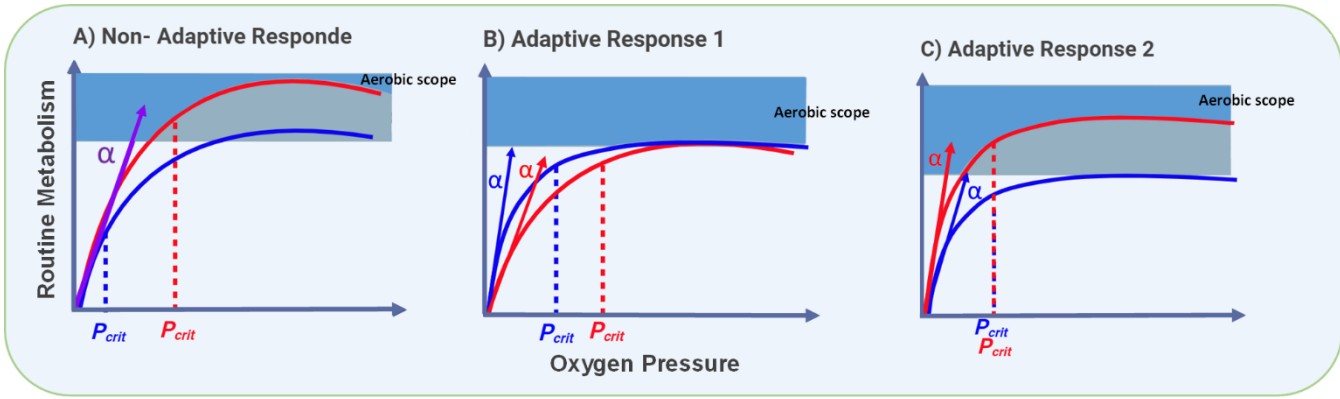

**Figure 4. The expected metabolic rate (MR) and critical partial pressure ($P_{crit}$) as a function of environmental oxygen**. The MR is theorized for two seasonal conditions: Autumn-winter (blue line) and spring-summer (red line) in a coastal upwelling zone, where $P_{crit}$ will move following changes in MR and the slope ($\alpha$) of the oxyconforming segment remains constant (A), an adaptive response 1 in which $P_{crit}$ increases upon a changing oxyconforming slope ($\alpha$) to maintain a constant MR (B), and an adaptive response 2, where $P_{crit}$ is maintained constant by increasing $\alpha$ during the upwelling season (C) (modified from Frederick et al., 2024). Here, the oxyconforming segment refers to the metabolic rate under oxygen levels

below P$_{crit}$, while the aerobic scope represents the range of MR between the basal and maximum metabolism for a given species.

Other adaptive responses include a shift to anaerobic metabolism when entering extremely low oxygen concentrations, such as those found in the core of the OMZ. This adaptation has been reported in actively migrating species, such as the krill *Euphausia* spp. (Riquelme-Bugueño et al., 2020). Several adaptive responses have already been described for zooplankton, inhabiting, or entering the OMZ. For instance, crustaceans inside the OMZ can experience long term physiological adjustments involving enhanced ventilatory capability, enlarged gill surfaces, shortened diffusion distances, and increasing respiratory proteins with high oxygen affinity (Childress and Seibel 1998). For example, in vertically migrating zooplankton entering the OMZ, Antezana (2002) observed that *Euphausia mucronata* had enlarged gill surfaces while actively respiring and swimming under depleted oxygen, indicating oxygen usage in the OMZ. In another study, Spicer et al. (1999) investigated anaerobic metabolism in the krill species *Meganyctiphanes norvegica* and found a significant increase in lactate concentrations as oxygen levels decreased to hypoxic conditions, allowing the species to withstand these conditions for extended periods. *Euphausia mucronata* has also been reported to have an elevated lactate dehydrogenase activity, enabling it to further tolerate prolonged exposure to hypoxic conditions (Gonzáles and Quiñones, 2002).

Finally, changes in behavior and distribution may reflect specific adaptive responses to hypoxia, as seen in many dominant zooplankton species avoiding the OMZ by restricting their vertical migration (Escribano et al., 2009; Tutasi and Escribano, 2019; Kiko et al., 2019). In the copepod *Acartia tonsa*, exposed to seasonal hypoxia driven by estuarine conditions (rather than the presence of an OMZ), avoidance of hypoxic bottom waters has been shown (Decker et al., 2003).

At ecosystem level, the different tolerances to low oxygen across species will determine their survival, and so leading tochanges in community structure and trophic webs. This not only due to mortality, but also due to changes in predator-prey interactions because of changes in abundance, migration, and habitat compression (Tutasi and Escribano, 2020). While several species will be negatively affected (including commercially exploited species), others more hypoxia-tolerant may expand their range of distribution, exploit new niches (Stramma et al., 2010), and therefore have access to new resources.

## 3 OXIDATIVE STRESS IN ZOOPLANKTON

Oxidative stress is a significant biological response to fluctuating oxygen levels, but it is rarely considered in marine environments. Driven by oxygen fluctuations, oxidative stress appears related to a state of respiratory imbalance in terms of O$_2$ uptake, supply and use, during which animals are unable to maintain constant tissue oxygenation/demand, and instead they experience rapid shifts between under-oxygenation and hyperoxygenation (Tremblay et al., 2010). This response is particularly

evident in organisms exposed to oxygen levels below their P$_{crit}$ values (as illustrated in Figure 4) and subsequently subjected to reoxygenation. Such conditions frequently occur in Eastern Boundary Upwelling Systems (EBUS), where oxygen levels can fluctuate dramatically from normoxia to hypoxia over short spatial and temporal scales (hours). Under such conditions, reactive oxygen species (ROS), as by-products of aerobic respiration are produced, and they play a critical role by signalling molecules. The production of ROS along with the release of the cytochrome c, AMP-activated protein kinase (AMPK),

mitochondrial DNA (mtDNA), and metabolites from the tricarboxylic acid (TCA) cycle, can all originate in the mitochondria (Martínez-Reyes and Chandel, 2022). Since all signalling molecules are highly dependent on the oxygen availability for mitochondrial functioning, their regulation is likely impaired by the exposure to variable oxygen levels near the OMZ.

    A higher production of ROS in the organisms may alter the DNA structure, result in modification of proteins and lipids, and trigger the activation of several stress-induced transcription factors (Birben et al., 2012). Available evidence suggests that

oxidative stress can generate a significant physiological cost in life expectancy, reproduction, the immune response, in addition to the effect on metabolism and growth (Zheng et al., 2021).

    Molecular oxygen (O$_2$) is the main biological electron acceptor, crucial in regulating cell functions. However, it is also the precursor of reactive oxygen species (ROS) formation because of normal cellular metabolism. The three major ROS of physiological significance are superoxide anion (O$_{2-}$), hydroxyl radical ($^-$OH), and hydrogen peroxide (H$_2$O$_2$) (Guérin et al.,

2001). The generation of these reactive oxygen species (ROS) has been extensively studied (Welker et al., 2013; Moreira et al, 2016; Giraud-Billoud et al, 2019). When animals are re-oxygenated after hypoxic exposure, ROS formation occurs, and if not neutralized by the body's antioxidant defences, may cause oxidative damage and eventually cellular disorder and death. ROS production is species-dependent which can further vary as a function of the intensity of hypoxia and exposure time to an oxygen-deficient habitat. However, ROS are not only produced by re-oxygenation after hypoxic exposure, in fact the ROS

production has also been reported to occur in hypoxic conditions for a variety of organisms, as reviewed by Hermes-Lima (2015). The exposure to hypoxia may lead to ROS production and eventually to production of antioxidant compounds as an adaptive response, a phenomenon described as preparation for oxidative stress (POS) (Hermes- Lima et al 1998., 2015; Moreira et al., 2016). Tremblay et al. (2010) showed that krill species adapted to hypoxia have a sufficiently high antioxidant protection whereas less adapted species suffered a strong oxidative stress measurable as lipid peroxidation. Thus, antioxidants play an

important role in neutralizing the oxidative action of free radicals.

    The mechanisms of cellular protection against ROS production include several antioxidant enzymes such as Superoxide dismutase (SOD), Catalase (CAT), Peroxidase, Glutathion S-transferase (GST), and Glutathione Peroxidase (GPx), and non-enzymatic such as Ascorbic acid (Vitamin C), Glutathione (GSH) Tocopherol and Carotenoids, where their role is critical in ROS detoxification processes in organisms subjected to oxidative stress. An imbalance between the level of ROS production

and antioxidant protection, it can result in oxidative damage to tissues and a state of oxidative stress (Birben et al., 2012).

    Linked to the presence of these radicals, a defence mechanism is the synthesis of antioxidant molecules capable of neutralizing the oxidative action of these free radicals. Several studies have described the presence of ROS production and corresponding antioxidant responses in zooplankton, summarized in Table 1.

**Table 1:** Zooplankton species exposed to different stress leading to ROS production. DVM (daily vertical migration) can be "Y" (yes migration) or "N" (non-migration) and "NI" (no information). Arrows in column 5 indicate the increase or decrease in the biomarker signal in relation at the stressor agent. * Indicate that the samples were obtained from areas having an Oxygen Minimum Zone.

| Group | Species | DVM | Stressor | Biomarker | Reference |
|---|---|---|---|---|---|
| Copepoda | *Acartia tonsa* | Y | Temp. (Heatwave) | ↑GST | von Weissenberg et al, 2021 |
| | *Acartia sp.* | Y | Temp/pH/DO | ↑GST | Glippa et al, 2018 |
| | *Calanus pacificus* | Y* | Temp/pH | ↑GST | Engström-Öst et al, 2019 |
| | *Limnocalanus macrurus* | Y | Contaminants | ↑SOD/↓LPX | Vuori et al 2015 |
| | *Eurytemora affinis* | Y | Temp/Sal | ↑GST | Cailleaud et al, 2007 |
| Euphausiacea | *Nictiphanes simplex* | Y* | Temp/Sal | ↑SOD/CAT/GST | Tremblay et al, 2010 |
| | *Nematocelis difficilis* | N* | Temp/Sal | ↑SOD/CAT/GST | Tremblay et al, 2010 |
| | *Euphausia eximia* | Y* | Temp/Sal | ↑SOD/CAT/GST/↓LPX | Tremblay et al, 2010 |
| Mysidae | *Neomysis awatschensis* | Y | DO | ↓SOD/↑↓CAT/LDH | Wang et al, 2021 |
| Pteropoda | *Limacina helicina* | Y* | Temp/pH | ↑GR/CAT/↓LPX | Engström-Öst et al, 2019 |
| Decapoda | *Scylla serrata* | N | Seasonal effect | ↑SOD/CAT/GPX | Kong et al, 2008 |
| Cephalopoda | *Sepiella maindroni* | NI | DO | ↑↓SOD/CAT/POD/LDH | Wang et al, 2008 |

**SOD**: Superoxide dismutase; **CAT**: Catalase; **GST**: Glutathione S-transferase; **LPX**: Lipid peroxidation;

**GPX**: Glutathione peroxidase; **LDH**: Lactato deshidrogenase; **POD**: Peroxidase; **GR**: Glutathione reductase


POS has been proposed as a mechanism to strengthen antioxidant defences (Hermes- Lima et al., 1998, 2015; Moreira et al., 2016). In the context of ROS associated with hypoxia, it is widely assumed that the antioxidant responses in a large variety of animals from marine, estuarine and freshwater systems (listed in Hermes-Lima et al., 2015) can be preceded by changes at the molecular and biochemical levels after exposure to variable oxygen levels from anoxia to hyperoxia, and such combined responses is therefore known as the POS phenomenon (Welker et al., 2013).

Considering the metabolic responses outlined in Fig. 4. The adaptive response 1 (Fig. 4B) is likely less stressful than the adaptive 2 and the non-adaptive response, from the oxidative stress perspective, as lacking large changes on aerobic metabolism (demands) should keep ROS production better regulated. The adaptive response 2 and the non-adaptive response (Fig. 4 A and C), instead, both demand a better regulation of antioxidant defences or POS, since changes on MR (changes in $O_2$ demand) and thus $O_2$ balance, will likely generate ROS at transitions between stable $O_2$ supply-demands states. Critical oxygen tensions ($P_{crits}$) measured in copepod species inhabiting the same area depicted in Fig. 3, in terms of oxygen variability, range from $2.77 \pm 0.89$ (*P.cf. indicus*) to $4.9 \pm 0.59$ kPa (*A. tonsa*), with *C. patagoniensis* in between with a $P_{crit}$ value of $3.83 \pm 1.89$ kPa (Frederick et al., 2024). Considering these values and the oxygen variability found in their environment (Fig. 3), only *P.cf. indicus* could maintain an aerobic metabolism stable if staying in the shallower 20 m strata, while the other two species will inevitably transit from aerobic to anaerobic metabolism in some seasons along the year. These metabolic transitions likely generate ROS and may also impact organisms performing diel vertical migration (DVM) as they traverse

water layers with varying dissolved oxygen levels, as illustrated in Fig. 3. Altogether, how this oxidative balance and POS operates in zooplankton inhabiting areas subjected to a shallow OMZ is an open question. The time required for developing

antioxidant defences also becomes an important issue in both migratory and non-migratory zooplankton, because of the short-time (few hours) within which animals are exposed to hypoxia-normoxia conditions during migration or due to the irregular pulses of upwelling causing the ascent or descent of the OMZ.  In the same context, and as mentioned above, most zooplankton avoid hypoxic waters by restricting their vertical distribution to the narrow and shallow normoxic layer in coastal waters, or by limiting the DVM avoiding entering the OMZ or at least the extremely low oxygen layer found at the OMZ core (Kiko and

Haus, 2019, Tutasi and Escribano, 2019). The highly variable DVM behaviour may thus play a key role for the adaptive response to hypoxia in the context of ROS production and POS processes.

Zooplankton performing extensive DVM can indeed enter the core of the OMZ, as described in several euphausiid species (Escribano et al., 2009; Riquelme-Bugueño et al., 2020). The incursions in extremely low oxygen waters and rapid re-oxygenation when ascending to near surface at nighttime may trigger ROS production and certainly antioxidant response. A

possible adaptive response to this periodic exposure to low or high levels of oxygen is possibility via POS by increasing ROS during the hypoxic phase of the DVM, as seen in Figure 5. In euphausiids that perform prolonged DVM, they are exposed to normoxic oxygen levels >200 µM in the photic zone, followed by hypoxia ($O_2 < 3$ µM). Their migration to deep zones (up to 200 m at dawn) takes less than 3 hours (Riquelme-Bugueño et al., 2020), which meansthey are exposed to a reduction/increase in oxygenation >30% per hour while swimming up or down.  Such stressful oxygen conditions over a short-time scale can

indeed trigger ROS production (Tremblay et al., 2010), and potentially POS during the nighttime phase (Fig. 5).

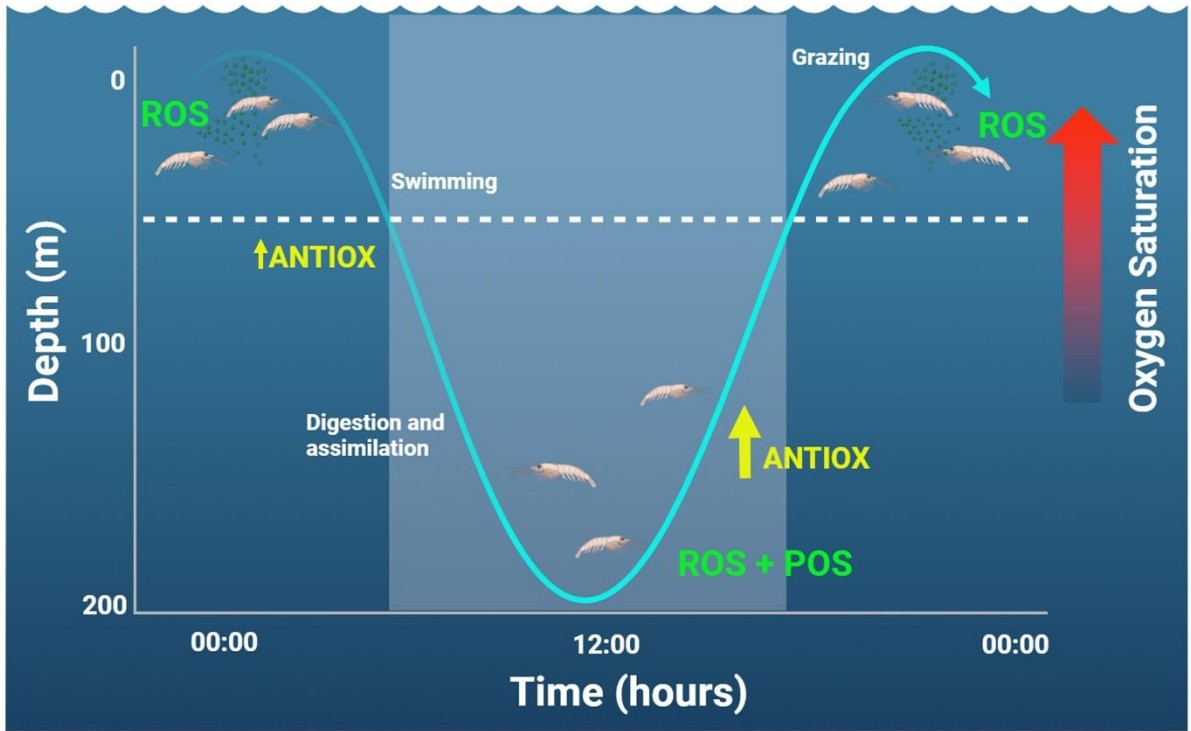

**Figure 5: Changes in the daily vertical migration and short-term response to oxidative stress in zooplankton.** Lighter
area shows the daytime. The vertical amplitude of migration represents that of euphausiids (Riquelme-Bugueño et al., 2020).
At nighttime an increase in the production of reactive oxidant species (ROS) is expected while ROS and the preparation for
an antioxidant response (POS) can take place during the daytime while animals stay below the photic zone (here represented
by the dashed line at about 50 m).

The DVM behaviour, considered as an adaptive response evolved to avoid visual predators during daylight conditions (Giraud-
Billoud et al., 2019), likely led to the adaptation of POS and hence allow a mechanism to mitigate the effects of ROS. Diel
cycles triggering antioxidant responses to ROS production have been shown to occur in planktonic organisms exposed to light-
dark conditions, such as *Daphnia pulex* (Cai et al., 2020), suggesting the existence of circadian rhythms of ROS production
and subsequent POS.  Several energy-intensive processes occur during DVM, such as grazing under oxygenated conditions,
followed by descent (increased activity) to digest ingested food (specific dynamic action) under hypoxic conditions. The
contribution of all these to ROS production is still unknown, whose production will likely mirror the gradients between
available oxygen at the environment and the metabolic demands.

However, not all zooplankton perform DVM. Copepods, for instance, significantly contribute to the bulk of zooplankton
biomass in the global ocean and remain within the upper 50 m of the water column (Escribano et al., 2009). These species,
however, may still be exposed to hypoxia when upwelling intensity increases, allowing the incursion of the upper limit of the
OMZ (<3 kPa $O_2$) into near-surface waters (Schneider et al., 2016). In case of strong upwelling, or extreme events driven by

the intrusion of mesoscale eddies into the coastal zone, they may also cause episodes of severe hypoxia in shallow waters and thus affect non-migratory copepods.

Over the annual cycle in EBUS, depending on latitude, the period of active upwelling has a strong seasonal variation in temperate areas or a weak and semi-permanent seasonal signal throughout the year in subtropical regions. For example, along the Chilean coast at mid-latitudes (30°–40° S), strong spring-summer southern winds drive an intense upwelling period, resulting in a shallow OMZ. In contrast, autumn-winter is characterized by reduced upwelling, a condition commonly referred to as the downwelling period. This seasonal variation can temporarily expose non-migrating zooplankton populations to

hypoxia during high frequency change (hours to days) in the upwelling season, and potentially triggering ROS production during the transitional period before of upwelling. The combined effects of seasonal upwelling conditions, vertical distribution of the OMZ, and lack of DVM behaviour within the upper 50 m layer, promoting ROS production and POS are illustrated in Fig. 6 for present and projected conditions. This figure illustrates a condition of downwelling (Autumn-Winter) with the upper limit of the OMZ below the photic zone where non-migratory zooplankton is dispersed and normoxic conditions prevail. This

oxic condition promotes the existence of ROS production, whereas strong upwelling and the shoaling of the OMZ into the photic zone can expose nonmigratory zooplankton to changing hypoxia-normoxia conditions, which result in POS+ROS.. The expansion of the OMZ may exacerbate this condition in the future of EBUS, by further compressing the vertical extent of the oxygenated habitat (Fig. 6).

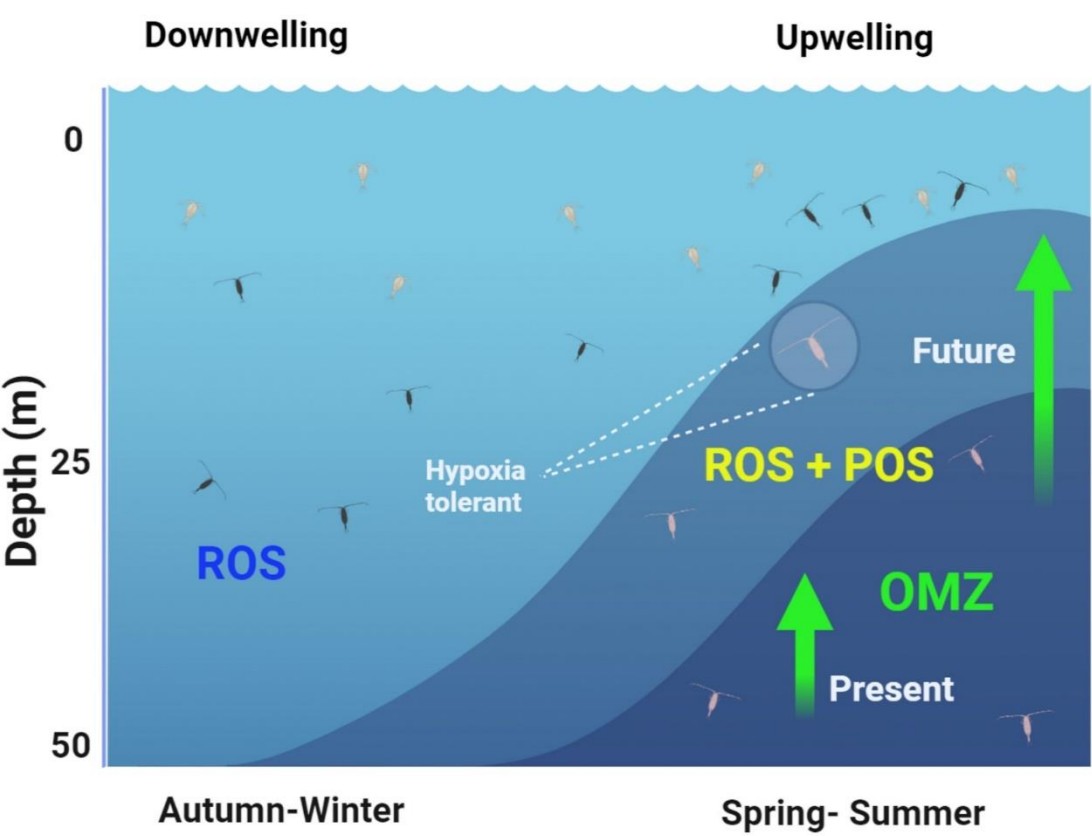


**Figure 6. The seasonal cycle of oxygen conditions of the photic zone (50 m) in EBUS at temperate regions**. Downwelling conditions prevail in the autumn-winter without presence of the oxygen minimum zone (OMZ) within the photic zone and so promoting oxidative stress (ROS). Contrasting conditions of spring-summer (active upwelling) characterized by presence of the OMZ promoting both ROS and the adaptive response (POS). Future condition may exacerbate the expansion of the OMZ

also compressing the oxygenated habitat for zooplankton.

It is challenging to carry out these studies in organisms inhabiting the water column, such as zooplankton, given the high variability in the physical-chemical parameters to which they are exposed. For example, planktonic copepods are very abundant, and they make up about 80% of the zooplankton biomass in EBUS, so it is imperative to further explore the effects

of changes in temperature, pH and dissolved oxygen (DO). An increase in antioxidant enzymatic activities indicates the activation of an antioxidant defence mechanism to deal with environment changes. This effect was reported by von Weissenberg et al. (2021) in *Acartia spp*. exposed to heatwaves, in experiments carried out at temperatures ranging between 9°C and 16°C. The authors observed a positive relationship between the increases in glutathione in response to increased environmental temperature and showed the deleterious effect over reproductive success during warming.

Regarding the dominant zooplankton in EBUS, a relevant issue to consider is the short life cycle of copepods (<2 months) and some numerically dominant euphausiids (<1 year). Under a seasonal upwelling regime, the exposure to hypoxia-normoxic may occur to different cohorts, and therefore the adaptive response (e.g. POS) might be seasonally adjusted. Although, if the OMZ continues its vertical expansion, there may not be sufficient time for developing adaptive responses with deleterious consequences for non-migratory populations, which comprise most upwelling inhabitant species.

## 3.1 NATURAL ANTIOXIDANT AGENTS FOR ZOOPLANKTON

In coastal upwelling environments, zooplankton must face extremely variable conditions, likely stressful for most organisms. In this very same environment, however, the presence of an exogenous source of antioxidants provided for example by diatoms might help mitigating the effect of oxidative stress due to such environmental fluctuations. Some studies show that planktonic diatoms have a high antioxidant potential (Goiris et al., 2012). Diatoms are rich in carotenoids, such as fucoxanthin and astaxanthin that play a crucial role in protecting against UV. Some diatoms such as *Skeletonema marinoi* and *Odontella aurita* can synthesize and accumulate ascorbic acid (Vitamin C) and phenolic compounds which also have antioxidant properties (Smerilli et al, 2019; Hemalatha et al, 2015).

The concentrations of antioxidants in diatoms are species-specific. For instance, Foo et al. (2017) quantified the total content of some antioxidants and their bioactive capacity in six diatom species and found that *Chaetoceros calcitrans* ($16.92 \pm 0.87$ mg TE. $g^{-1}$ DW) and *Isochrysis galbana* ($21.55 \pm 1.58$ mg TE. $g^{-1}$ DW) showed the highest antioxidant activity, followed by *Odontella sinens* and *Skeletonema costatum* which exhibited moderate bioactivity (<13 mg TE. $g^{-1}$ DW), while *Phaeodactylum tricornutum* and *Saccharina japonica* displayed the lowest antioxidant activity (<5 mg TE. $g^{-1}$ DW) among the examined algae species. Most of these diatoms dominate the upwelling zone, specially *Chaetoceros, Odontella* and *Skeletonema*, during the upwelling season (spring-summer) (Anabalón et al., 2007), and they are an important item of the zooplankton diet (Vargas et al., 2006).

## 4 ECOLOGICAL CONSEQUENCES

Ocean deoxygenation and the loss of oxygen in EBUS is an ongoing phenomenon, and planktonic organisms must inevitably cope with this gradually increased hypoxia, (on average) and a changing dynamic of the ocean (change in occurrence and extreme events). The ecological consequences are far from understood and will largely depend on the ability of zooplankton to strengthen their capacity to tolerate mild or severe hypoxia, exploit plasticity and maternal effects to their maximum, or to develop new adaptations. In any case, these responses may likely come at some cost and likely with trade-offs on other metabolic/energy-demanding processes. The outcome from the new demands implies a reduction in energy otherwise available for growth, feeding and reproduction with further consequences in the population dynamics. The high dynamism of the water

column in EBUS, with factors such as light, oxygen, $CO_2$, temperature, and food, along with zooplankton processes like feeding, DVM, and digestion, makes it challenging to identify the most challenging water strata, especially when considering temporal variation. Hypoxia conditions can also lead to changes in behaviour (upon stress) and spatial distribution and so altering for example prey-predator interactions, although such consequences are difficult to predict. The pelagic food-web and community structure will likely be affected with biogeochemical consequences in the context of the C and N recycling and ecosystem productivity. A conceptual model illustrating how variable oxygen levels in the water column can modulate the metabolic rates, drive their adaptive modes, and exert ecological consequences at population, community and ecosystem levels in zooplankton inhabiting the upwelling zone of EBUS, is shown in Fig. 7. Further consequences driven by potential oxidative stress (ROS) are also included (Fig. 7).

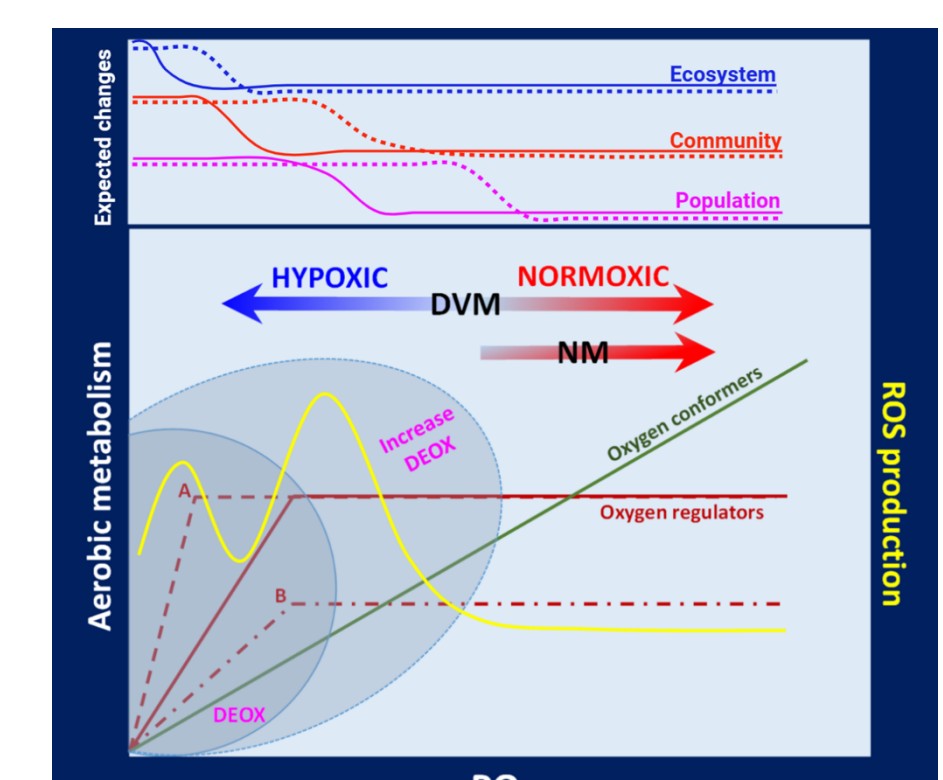

**Figure 7. Conceptual model of aerobic metabolism in zooplankton.** This scheme illustrates the different adaptive modes of zooplankton when responding to variable oxygenation (hypoxic to normoxic conditions) in the water column of EBUS. Upon variable oxygen pressure ($PO_2$) the aerobic metabolism of oxygen conformers and oxygen regulators is shown. Oxygen regulators may exhibit two types of response; A: changing $P_{crit}$ but maintaining a constant metabolic rate (MR), or B:

maintaining a constant P$_{crit}$ but modifying MR. Reducing P$_{crit}$ maintaining a constant MR (Adaptive mode A) may confer a
375 greater capacity to withstand severe hypoxia satisfying the metabolic demands, while adaptive mode B implies a better
exploitation of favourable conditions at expenses of decreasing aerobic scope while maintaining a constant P$_{crit}$. These
metabolic responses can be further exacerbated under increasing deoxygenation in the upwelling zone (grey areas). Individuals
performing DVM between hypoxic-normoxic conditions are more likely exposed to oxidative stress (yellow line) as being
subjected to strong fluctuations in PO$_2$ in the proximity of P$_{crit}$. At population, community and ecosystem levels, ecological
changes are expected to increase with hypoxic conditions but exhibiting a lag among organization levels. These changes can
go from altered abundances and biomass in populations, altered community structure and modified whole food web in the
ecosystem, Changes should become stronger under increased deoxygenation (dotted lines).

## 5. REMARKS AND RECOMMENDATIONS

Ocean deoxygenation will potentially increase the in coming decades and the marine aerobic biota will gradually be exposed
to more stressing conditions in terms of oxygen levels. In EBUS, zooplankton is expected to perhaps better cope with increasing
hypoxia, considering their high plasticity and the permanent presence of the OMZ to which different adaptations have already
been evolved. However, the extent to which increasing hypoxia can impact specific populations in the upwelling zone requires
attention since potential alterations at the base of the food web can have further consequences in the whole marine ecosystem.
In this context, the assessment of plankton community structure through time series observations in the upwelling zone
constitutes the most suitable proxy to examine the community responses to ongoing deoxygenation, and so long-term time
series are extremely valuable for accurate predictions. Also, the use of molecular methods to examine how individuals can
modify their gene expression to cope with hypoxia, and eventually activate antioxidant responses, are also necessary
approaches when aiming to the understand and predict the ecological consequences upon the expected severe conditions of an
395 oxygen-deprived water column. New molecular methods, such as proteomic, which may recognize specific enzymatic systems
actively participating in metabolic processes and biosynthesis of antioxidant compounds, can offer novel lines of research to
understand adaptive and non-adaptive (phenotypic plasticity) responses to increasing hypoxia at the molecular level.

**COMPETING INTEREST**

The contact author has declared that none of the authors has any competing interests.

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
