# Peer review of "Reviews and synthesis on increasing hypoxia in eastern boundary upwelling systems: zooplankton under metabolic stress"

_EGUsphere, 2024_

## Author Comment (AC1)

Reply to reviewer's comments

This is an interesting paper that extends some of the results presented in Frederick et al. (2024) regarding adaption of the metabolic rate of copepods to low oxygen conditions in an eastern boundary upwelling system (EBUS). While the title and abstract imply that this manuscript will be a general review of hypoxia stress on EBUS zooplankton, their focus is on modification of respiration (Pcrit) and reactive oxygen stress (ROS). There are other zooplankton individual, population and community responses such as reductions in fecundity, growth rate, species size, changes in the species composition, etc. which are not addressed in this Review and Synthesis. The authors might want to acknowledge this narrower focus (perhaps change the Title) and provide references to other reviews of hypoxia effects on zooplankton.

**R. We appreciate comments and suggestions from the reviewer. It is true that previous studies have reported different aspects of ecological effects of hypoxia on plankton. In our review, we are not intending to cover all these aspects, but to focus specifically on some physiological aspects largely less considered, and which we believe are essential to understand how the organisms may or may not adapt to cope with potential future ocean deoxygenation. In this regard, we agreed with the reviewer that is better to change the tittle a bit to be more specific. Our new tittle will be "Reviews and synthesis on increasing hypoxia in eastern boundary upwelling systems: zooplankton under metabolic stress"**

Line 84: The authors cite Chisholm and Rolf (1990) as the reference for oxygen-regulators/conformers as related to Pcrit. This citation was not in the references and I do not think it is the proper reference for this topic.

**R. We agree with this comment and it was a mistake to add this cite. We are now correcting and the right cite is Portner & Grieshaber (1993) which has also been added to the reference list.**

The focus of the paper is on zooplankton of the EBUS, most of which are copepods which do not have gills. Many of the references (lines 95-98) provided on oxygen regulation are for fish and invertebrates which have gills which may have different regulatory oxygen capacity than copepods which obtain oxygen by diffusion through their body surface.

**R. The reviewer is right, although the modelling of these metabolic responses was indeed developed in fishes and then lately applied to invertebrate such as crustacean including**

**copepods without gills. We have modified the paragraph to be more specific and now citing other studies on copepods.**

The Legend for Figure 3 is not correct. The Y axis is Metabolic Rate (presumably oxygen consumption/zooplankton and the X axis is Oxygen Partial Pressure (presumably in kPA). Pcrit is the oxygen partial pressure at which the slope of the line changes.

**R. Thanks for spotting this, it has now been amended**

Line 127-128. It should be noted that the study referenced for Acartia tonsa was conducted in Chesapeake Bay, a shallow estuary in the U.S. not an OMZ.

**R. Agreed. The text has been modified. We now report these effects on other copepod species, and in the case of Acartia we now specify that such hypoxic conditions have an origin other than due to the presence of an OMZ.**

Line 145. Perhaps a reference should be listed to support this sentence.

**R. Agreed: a reference has been added (Zheng, 2021 for review)**

---

## Author Comment (AC2)

**eastern boundary upwelling systems: a major stressor for zooplankton'
(Frederick et al)**

**General comments**

This project aims to provide a synthesis of zooplankton responses to changes in hypoxia in Eastern Boundary Upwelling Systems (EBUS). To this end, the authors motivate their review by discussing changes in the biogeochemistry of the EBUS due to climate change, with a particular focus on upwelling and the vertical expansion of hypoxic layers and the Oxygen Minimum Zone (OMZ). The authors then introduce two different adaptive modes by which organisms change their metabolic activity to decreasing oxygen concentration and show possible adaptive responses, as well as the non-adaptive response. This is then followed by an extended section on the oxidative stress in zooplankton, where the authors explore the consequences of hypoxia and the production of reactive oxygen species for both migratory and non-migratory zooplankton.

This review is very timely as ocean deoxygenation is increasing globally. However, in my opinion, the review could go more into detail with respect to the mechanisms presented, and provide more details on the individual- and population-level consequences of an increase in the exposure of zooplankton to low oxygen levels.

**R. We thank the reviewer for his(her) thoughtful comments and the very detailed revision. We agree on that the issues raised need to be addressed**

Furthermore, the link between the adaptive responses and the oxidative stress was for me not entirely clear. In other words, It was unclear to me if the responses to oxidative stress change depending on the adaptive responses shown in Figure 3, and whether one response would be favored over another under increasing deoxygenation.

**R. Thanks for this comment, inviting to further speculate on the oxidative stress consequences of the metabolic responses identified. We agree that the link between metabolic adaptation to variable oxygenation and the effect of oxidative stress (ROS) on this adaptive response is not so evident in our review and needs to be clarified. Indeed, the proposed mechanisms for metabolic adaptation to cope with changing oxygen may operate within a range of hypoxia above Pcrit, and below this the ROS can be triggered upon later re-oxygenation. However, this possibility was not clearly represented in the MS, and therefore in the section in which we describe ROS and POS we**

**are now recalling this Fig. 3 (now Fig. 4) and discuss on how these metabolic adaptations can relate to them. The actual paragraph is:**

**"An important biological response linked to variable oxygen levels, and rarely considered in the ocean, is oxidative stress. The phenomenon can occur because the variations in oxygen levels may range from normoxia to hypoxia at short spatial and temporal scales (hours) in some areas, such as in EBUS. Driven by such fluctuations, the oxidative stress appears related to a state of respiratory imbalance in terms of $O_2$ uptake, delivery, and usage, during which the animals cannot maintain a constant tissue oxygenation and, instead, undergo rapid changes between under-oxygenation and hyper-oxygenation (Tremblay et al., 2010). This can indeed occur under stressful conditions in individuals subject to oxygen levels below their Pcrit values (as shown in Fig. 4), and which thereafter undergo re-oxygenation. Therefore, as a product of aerobic respiration, the production of reactive oxygen species (ROS) can occur."**

Finally, I felt that some of the physical and behavioral processes presented in the review are oversimplified or do not match other findings/definitions. Here I am referring to the oversimplification of increased wind-driven upwelling across the EBUS and the depth of diel vertical migration (DVM). I think the authors oversimplify the processes by which upwelling has changed across the EBUS, where for instance the review by Bograd et al. (2023) does a great job at detailing the differences in upwelling intensity across different EBUS. With respect to the DVM, the authors use a euphotic zone of 50 m rather than the usual 200 m, and the vertical extent of DVM shown seems extremely shallow, which does not match the description in the text.

**R. We appreciate this comment and suggestions. We agree on that changes in upwelling in EBUS forced by global warming varies strongly spatially and so having influence on both vertical and horizontal distribution of the OMZ, possibly impacting DVM too. We considered 50 m as a prevailing photic zone in coastal waters more directly influenced by upwelling and presence of the OMZ, and where most zooplankton concentrate. Although the boundaries of the OMZ on oceanic waters must consider a deeper photic zone (likely between 100 and 200 m deep). We also agree that some zooplankton, such as euphaudiids can migrate deeper than w 200 m, entering the core of the OMZ. In this respect we have modified Fig. 1 to represent a cross-shelf variable depth of the photic zone and extend the discussion on how a spatially heterogeneous upwelling response to global warming can affect the OMZ distribution in different EBUS and over the meridional gradient. We have**

**also added vertical arrows to represent a variable DVM amplitude depending on the position of the upper boundary of the OMZ, although indicating in the text that some zooplankton (e.g. euphausiids) may override this potential barrier to perform DVM. Certainly, we are now citing the review of Bograd et al. (2023), as well as other important works, such as Xiu et al. (2018), Schneider et al. (2016). The most important modifications in Figures and text are detailed below.**

**Specific comments**

L.15: As mentioned above, I think the authors should address the uncertainty in upwelling trends depending on the data set, region, and proxy variable used, since this seems to be the one of the key process in the paper by which organisms can be exposed to low oxygen concentration. For instance, regional observations have shown a weakening or no significant trend in upwelling intensity for the Canary Upwelling System (Bode et al. 2019). Or Pardo et al. (2011) that found a small weakening trend in the Humboldt System and no trend in the California Current System using trends in sea surface temperature.

**R. We thank for the suggestions and references. We are modifying the text saying that this issue may vary depending on projections for different EBUS and there is still debate and uncertainty on the future of upwelling. There are several works suggesting increasing upwelling in EBUS, though it seems that increasing upwelling may mostly occur at mid-latitudes (Wang et al. 2015, Xiu et al., 2018), including the Benguela EBUS (Santos et al. 2012), the Chilean EBUS (Schneider et al. 2016), although there is also no evidence of increasing upwelling in other EBUS (Pardo et al. 2011, Bode et al., 2019). We are now including these references to discuss the issue related to the uncertain future upwelling in EBUS. We have also modified Fig. 1 to illustrate that increasing upwelling is a possibility, but still uncertain. This is New Fig. 1 and its caption:**

[Figure]

Figure 1. Projected effects of expansion of the OMZ in eastern boundary upwelling systems (EBUS). Upper panel) Under present (initial) conditions, wind-driven upwelling rises the OMZ system and brings cold-water into shallow depths at the inshore as illustrated by the 12°C isotherm, and so fertilizes the photic zone and promotes plankton aggregation. Lower panel) Ocean warming effects manifest mainly at surface in the offshore region, while a vertically expanded OMZ along with an eventual (although uncertain) increase in upwelling may cool down the coastal zone and further shoaling the OMZ at the inshore area, causing more hypoxia and vertically reducing the oxygenated habitat. Vertical compression of the oxygenated habitat may restrict the vertical distribution and diel vertical migration (DVM) of zooplankton.

Also, the modified paragraph is now:

"In some areas, mainly at mid-latitudes, of the four major eastern boundary current systems (EBUS) (Chavez and Messié, 2009), the effect of climate change has been associated with an intensification of the physical forcings driving coastal upwelling (Bakun et al., 2010; Xiu et al. 2018, Bograd et al., 2023), leading to several changes on the physical-chemical properties of the water column, including a gradual cooling in the last few decades (Santos et al., 2012; Schneider et al., 2016). However, other studies have found no evidence of increasing upwelling or trends in upwelling intensity, as based on time series observations for several decades in same EBUS (e.g. Pardo et al., 2011; Bode et al., 2019). Trends of upwelling intensity in EBUS is therefore still matter of controversy, and the predictive models reveal much uncertainty on future upwelling regarding its spatial and temporal variability (Bograd et al. 2023). Upon potential increase of

**upwelling, favourable winds in EBUS bring colder water and more frequent occurrences of upwelling events (Breitburg et al., 2018), although some modelling work also suggests an extension of the upwelling period and spatial homogenization of upwelling over the alongshore axis (Wang et al., 2015). Stronger upwelling is ultimately thought to be a response to the strengthening of large-scale pressure gradients linked to global-scale climate change (Garcia-Reyes and Largier, 2009). With the intensification of the coastal upwelling, a shoaling of the oxygen minimum zones (OMZ) in coastal waters may take place and so compressing the upper highly oxygenated layer (Khön et al., 2022). The closely linked effects of increasing upwelling, cooling of the water column and shoaling of the OMZ in EBUS driven by global warming are illustrated in Fig. 1..”**

L.29-31: This sentence contains a strange redundancy stating that a warmer ocean drives increases in mean global sea surface temperature. In addition, I would argue that the warming drives the physico-chemical changes rather than just a warmer ocean. Please revise the sentence.

**R. Agree. We have modified this paragraph to avoid redundancy. Now it should read "climate change drives several physical processes....."**

L.34: Here, I suggest the authors be more quantitative and say by how much the oxygen concentration has changed since the middle of the 20th century. With such a measure, the authors can compare the long-term changes with seasonal and spatial changes in oxygen concentration.

**R. we have added an estimated reduction in oxygenation in about 2% of the global ocean with corresponding cites.**

L.40-43: Given that short-term changes in oxygen are of interest for this review, I suggest to also include the role of extreme events, which can lead to transient habitat reductions of variable duration in EBUS such as the California Current System and the Humboldt System (Köhn et al., 2022).

**R. We are now referring to such extreme events as potential perturbations related to OMZ distribution and hypoxia and their potential effects of zooplankton, citing the work of Köhn et al. (2022). The added paragraph is:**

**" In some cases, the minimum oxygen concentrations in the OMZ cores have also been further reduced, intensifying the OMZ (Chan et al., 2008). Increasing hypoxia driven by these trends in loss of oxygen can be further exacerbated by extreme events caused by the action of mesoscale eddies producing intense episodes of hypoxia in the upwelling zone (Khön et al., 2022)."**

L.47-48: As spatial heterogeneity in oxygen conditions is one of the key factors determining the exposure of zooplankton to low oxygen, I urge the authors to explore how the spatial changes in upwelling due to the poleward displacement of coastal upwelling winds (Rykaczewski et al. 2015) has or will likely change the overall oxygen conditions in the poleward vs the equatorward boundaries of EBUS.

**R. Although there is much uncertainty and debate on the future of upwelling in EBUS (Bograd et al. 2023), some modelling suggests a potential extension of the upwelling period and alongshore spatial homogenization of upwelling. Considering these possibilities the level of oxygenation for zooplankton may not change vertically, but perhaps also horizontally alongshore and cross-shelf as well due to the OMZ expansion (Grégorie et al., 2021). This implies spatial variation in the oxygenated habitat with ecological consequences for zooplankton. In this respect, we are now describing these modelling results and their consequences in the Introduction and Discussion of the Review. The paragraph being modified now reads like:**

**"Upon potential increase of upwelling, favourable winds in EBUS bring colder water and more frequent occurrences of upwelling events (Breitburg et al., 2018), although some modelling work also suggests an extension of the upwelling period and spatial homogenization of upwelling over the alongshore axis (Wang et al., 2015)."**

L.54-55: Here the authors should provide some examples of the various ecological and biogeochemical consequences, preferably with a quantitative metric, as this would add more meaning to their statement and improve their review for future readers.

**R. We are now adding several examples on some ecological consequences of hypoxia. The paragraph is:**

**"The ongoing combined processes, deoxygenation, increasing upwelling, OMZ expansion and potential latitudinal expansion of upwelling will alter the oxygen conditions in upper layers (<50 m) in EBUS, where plankton becomes concentrated, with various ecological and biogeochemical consequences. In this respect, Ekau et al. (2010) demonstrated that hypoxic conditions can alter the zooplankton community composition in the Benguela EBUS. This occurs because of variable tolerance to hypoxia in some distinctive groups, being euphausiids for example better adapted to low oxygen (<0.1 mL/L) compared to**

copepods. Escribano et al. (2009) also described a strong vertical zonation of zooplankton depending on variable tolerance to hypoxia in the northern upwelling zone of Chile. Variable tolerance to hypoxia is also reflected in some species-dependent physiological rates of copepods, as found in the calanoids *A. tonsa* and *C. chilensis* whose egg production rate and hatching success were strongly positively correlated to oxygen concentration under laboratory conditions (Ruz et al., 2015). In the same context, not only vertical distribution, but also the vertical amplitude of the diel vertical migration can also be strongly modulated by hypoxic conditions forced by position of the OMZ core and its upper boundary (Tutasi and Escribano, 2020; Riquelme-Bugueño et al., 2020). "

L.56: The authors introduce the terms normoxia and mild or severe hypoxia. I understand that these levels are species-specific, however, it would help to have a range or a mean of a selected number of species. I would also imagine that using the levels for copepods would suffice given their abundance and importance.

R. Agree. We are adding a paragraph to define these terms. This is:

"Aerobic metazooplankton inhabiting the upwelling zone is thus expected to be exposed to variable levels of oxygenation from normoxia to mild or severe hypoxia, depending on their distribution and migrating behaviour, but also depending on variable levels of tolerance to hypoxia. In copepods, which represent the major contributors to zooplankton biomass in EBUS, oxygen levels 1-2 mL L$^{-1}$ may represent mild hypoxia, while concentrations <1 mL L$^{-1}$ should be considered as severe hypoxia for most species (Wishner et al., 2018; Frederick et al., 2024)."

L.79-85: The authors explain very clearly two adaptive modes and illustrate the differences between them. To increase the impact of the review, they should also mention characteristic timescales at which such changes can occur in zooplankton and provide examples of possible organisms that follow each adaptive mode if possible.

R. Agree. We are now providing the timescales over which these adaptive responses can take place for vertically migrant and non-migrant zooplankton. To better illustrate this, we are also adding a new Figure (new Fig.3) in which we illustrate the oxygen gradient in the upwelling zone from a database of oxygen of a time series at the upwelling zone off central-southern Chile. The new paragraph and Figure are:

**"The adaptive responses to variable levels of oxygen illustrated in Fig. 2 may take place over short-term timescales (hours), as driven by exposure to a vertical gradient of oxygen in the water column when performing DVM in migrating species, or in the case of non-migrating species due to vertical mixing in the water column as forced by upwelling pulses or changes in wind conditions promoting mixing. Both migrating and non-migrating species can thus be exposed to variability in oxygenation in the upwelling zone which is characterized by a marked oxygen-stratified water column (Fig. 3). Clearly within the photic zone (about 50 m in the coastal zone) the annual cycle of oxygen conditions reveals the existence of normoxia, mild and severe hypoxia habitats with which vertical migrant and non-migrant zooplankton must cope with depending on their vertical distribution in this layer (Fig. 3)."**

[Figure]

**New Fig. 3. The annual cycle (monthly climatology) of oxygen concentration in three strata in the upper 50 m layer at central-southern Chile (36°30 S). Oxygen data are from the time series study at Station 18 off central-southern Chile during the period 2002-2016 (Frederick et al., 2024).**

Furthermore, given the evidence from Cobbs and Alexander (2018) on the existence of other response types to progressive hypoxia among marine animals; including zooplankton; the authors could expand their review to include more response types.

**R. We thank for this comment. Our proposed model based on two types of metabolic responses to variable oxygenation is the traditional view, based on studies on a variety of organisms. The analysis provided by Cobbs and Alexander (2018) mostly describes potential empirical results, although as they state these results do not necessary can reflect a functional or adaptive response. In any case, we agree that is an important issue to consider,**

**because we have seen that the evolution of respiration as a function of oxygen is indeed very variable and not easy to interpret or to fit to single mathematical functions. For examples, in situations with animals under stress regulation or no regulation can occur or having different phases during the response. We are now discussing this issue in the text and including this reference.**

L.99-103: The authors explain very briefly how maternal effects can play a role in shaping the plasticity of stress responses in organisms, putting a focus on the effects of life-history on individual-level responses and possible consequences for the population. However, the authors then remain relatively vague in explaining how this mechanism works and do not explain possible feedbacks and consequences on the population.

**R. We agree with reviewer in the need for extending and clarifying this issue. Since copepods are rather short-lived animals (<months), we hypothesize that even though maternal effects can be passed on to seasonal cohorts, allowing the offspring inherit adaptive characters to cope with variables conditions, this offspring can also react to new upcoming conditions by activating some genes coding for proteins developed for physiological or biogeochemical functions for improved fitness. We have no data or evidence for this, although many works have evidenced the environmental effects on gene expression (epigenetics). We are now extending this paragraph to propose such mechanism as a hypothesis that needs consideration in future studies.**

L.125-129: The authors name some changes in behavior and distribution as a result of exposure to hypoxia. While I agree with the examples in the text, it would help to give other examples of strategies observed in the field (e.g. Hauss et al., 2016).

**R. We are now adding more examples on responses to low-oxygen conditions, such as those provided by Hauss et al. (2016) who reported various responses to a shallow OMZ associated to a subsurface eddy in the northeast tropical Atlantic, depending on the different groups of zooplankton. We have also found aggregations of zooplankton just at the base of the oxycline to avoid predation, but also avoiding extremely low-oxygen water (Donoso and Escribano, 2014). We are also adding the migration response of some copepods to the lower boundary of the OMZ, such as that reported by Wishner et al. (2008).**

L.132-133: Here again, it would be important to have a sense of the spatio-temporal scales to fully grasp how the exposure to low oxygen conditions changes along the life-history of individual organisms.

**R. Agree. We are now describing that processes of interaction between copepods and oxygen gradients controlled by distribution of the OMZ may occurs across short-term time scales (hours) due to vertical migration and some physical processes controlled by vertical mixing and upwelling pulses, also over microscale and mesoscale spatial scales due to oxygen gradients occurring within the photic zone and across physical structures, such as mesoscale eddies and fronts.**

L.143-145: Here, it would be important to get some comparison of the effect of oxidative stress on the different physiological costs, or rather how was the significance measured?

**R. We are now expanding the issues related to consequences of ROS on physiological and metabolic changes. For example, in addition to changes in behavior (such as avoiding hypoxia), stressful conditions may result in reduced activity (slow swimming, reduced vertical migration), as reported in Euphausiids (Tremblay et al., 2010) and reduced enzimatic activity in copepods Glippa et al. (2018).**

L.153-154: The authors explain how ROS production occurs both during hypoxic conditions and after re-oxygenation. Thus my question was what are the implications of this, or can you say something about the difference in ROS production under the different conditions and by how much it changes?

**R. Here, the issue is that ROS will always occur under oxygenated conditions and therefore animals are producing antioxidant responses. However, an individual exposed to hypoxia with lower tolerance to low oxygen is subject to a stressful condition which will trigger ROS when reoxygenating, although as we also said ROS can been occur during the exposure to hypoxia. However, the most important point here is being subject to an extremely variable oxygen condition, and so inducing POS and ROS with metabolic costs. We are now modifying some of the text to clarify this statement.**

L.179: Can you say something about the consequences of a limited DVM and compare it to other strategies observed in the field (see Hauss et al., 2016)?

**R. Reduced DVM may have implications for populations and with some biogeochemical consequences. For example, increasing mortality from predation, promoting other biological interactions from more aggregation, and limiting the vertical fluxes of C and N mediated by active transport (Steinberg and Landry, 2017).**

L.203-204: The authors explain how hypoxic conditions may reach the surface due to strong upwelling. Here, it would be important to compare the exposure of such surface conditions to what migratory zooplankton experience during DVM. Maybe you could provide some characteristic intensities or durations of exposure?

**R. Upwelling pulses prevail over a time scale of a few days, and therefore these events can have more drastic consequences than exposure to oxygen gradients during hours by DVM. Hypoxia driven by such upwelling events may affect the entire near-surface community, while exposure to sharp oxygen gradients can affect just to vertically migratory populations. Hypoxia events have been reported causing massive mortalities in some pelagic organisms inhabiting the upper mixed layer. We are now expanding the discussion on the time and spatial scales over which zooplankton may become exposed to hypoxia.**

**Technical corrections**

L.37: What do you mean by "becomes even more critical"? Do you mean deoxygenation is enhanced or that it is more critical for marine life?

**R. We meant for marine life, so we have modified to sentence to better state this.**

L.54: It should be become rather than becomes. In addition, what is meant by "plankton become more concentrated"? Do you mean they are more abundant? There is also a similar phrasing in the caption of Figure 1.

**R. Ok. Corrected. We mean more abundant. Corrected now**

L.74: It should be "At the ecosystem level…". This sentence is also rather convoluted. Can you rephrase it for clarity as it has many clauses.

**R. Corrected**

L.84: The reference Chisholm and Roff, 1990 is not in the bibliography.

**R. This reference was removed from the text**

L.85: Please introduce the abbreviations used on the first mention. Here you can introduce the abbreviation for metabolic rate (MR).

**R. Agree. Done**

L.89-91: Please split up this sentence to increase readability.

**R. Agree. Done**

Figure 2: The term routine metabolism was not introduced in the main text. In addition, the axis descriptors do not match the text descriptors, making the interpretation of the figure more difficult. The caption should include more information of what is shown in the figure. Finally, unless I missed something, I was not able to find a similar figure in Rogers et al., 2016 from which figure 2 was adapted.

**R. We are now defining routine metabolism and better describing this Figure 2. The figure was indeed based on Figure 1 of Rogers et al. 2016. We are now specifying that is based on, but not modified from**

L.97: The authors introduce the term "maximum oxygen supply capacity" without definition. As this is a review, I feel that important terms like this should be defined.

**R. Agree this term is now defined.**

L.137: Here, a lot of abbreviations are used without introducing them first.

**R. We have revised all acronyms and symbols to introduce them.**

Table 1: Some abbreviations are not introduced (e.g., GR in Pteropoda). Also, what is the difference between having different biomarkers linked by a slash and a dash? In the caption, to the best of my knowledge it should be "Lactato deshidrogenase" and "Peroxidase".

**R. We have revised all acronyms and symbols to introduce them.**

L.173-174: Can you say in which organisms POS has been proposed as a mechanism to strengthen antioxidant defences?

**R. We are now citing examples provided in Hermes-Lima et al. (2015).**

L.175: To the best of my knowledge, it should be migratory and non-migratory.

**R. Agree. We have corrected to text.**

L.185: The abbreviation POS seems strange because "doing POS" would mean "doing preparation for oxidative stress" rather than preparing for oxidative stress.

**R. Agree correction done.**

L.186: Should read "The interplay between ROS production and POS...". Please revise throughout the manuscript when using abbreviations (e.g. L 190).

**R. Corrected**

L.189: Should read "reduction/increase"

**R. Corrected**

Figure 4: The depth axis does not match the reference "Riquelme-Bugueño et al. (2020)" or the main text. The abbreviations used in the figure are not explained. It might be better to show the direction of change with arrows rather than with multiple "+" signs, as this would resemble the symbols used in Table 1. Also, what is the meaning of the dotted line.

**R. Corrected now**

**New Figure 4**

[Figure]

L.209: What is meant by "during high frequency change"?

**R. We meant during high frequency pulses of upwelling. Corrected now**

Figure 5: As the main text refers to seasonal changes, I was expecting to see a timeseries (resembling Figure 4). Also, the abbreviations are not properly explained in the caption.

**R. Fig. 5 (now Fig 6) has been modified. This is new Figure:**

[Figure]

L.227: Rather than saying it seems more difficult, you can simply say that it is more difficult.

**R. Agree. Corrected**

L.232: DO has not been introduced.

**R. Corrected**

L.232: Remove "front".

**R. Done**

L.234: Rephrase to "experiments carried out at temperatures ranging between 9°C and 16°C."

**R. Thanks. Done**

Section 3.1: I was initially confused as this section begins with explaining the high antioxidant potential of diatoms in a zooplankton paper. It wasn't until the last paragraph in this section that I realized how this fits in the story line. Thus I would suggest changing the sequence in which the information is presented to increase readability.

**R. Thanks for the suggestions, we have modified the text following your recommendation.**

**Thanks for the references, they are all very important for the review.**

**Bibliography**

Bode A, Álvarez M, Ruíz-Villarreal M, Varela MM. 2019. Changes in phytoplankton  production and upwelling intensity off A Coruña (NW Spain) for the last 28 years. Ocean Dyn. 69:861–73

Bograd SJ, Jacox MG, Hazen EL, Lovecchio E, Montes I, Pozo Buil M, Shannon LJ, Sydeman WJ, Rykaczewski RR. Climate Change Impacts on Eastern Boundary Upwelling Systems. Ann Rev Mar Sci. 2023 Jan 16;15:303-328. doi: 10.1146/annurev-marine-032122-021945. Epub 2022 Jul 18. PMID: 35850490.

Cobbs GA, Alexander JE Jr (2018) Assessment of oxygen consumption in response to progressive hypoxia. PLoS ONE 13(12): e0208836. https://doi.org/10.1371/journal.pone.0208836

Hauss, H., Christiansen, S., Schütte, F., Kiko, R., Edvam Lima, M., Rodrigues, E., Karstensen, J., Löscher, C. R., Körtzinger, A., and Fiedler, B.: Dead zone or oasis in the open ocean? Zooplankton distribution and migration in low-oxygen modewater eddies, Biogeosciences, 13, 1977–1989, https://doi.org/10.5194/bg-13-1977-2016, 2016.

Köhn, E. E., Münnich, M., Vogt, M., Desmet, F., & Gruber, N. (2022). Strong habitat compression by extreme shoaling events of hypoxic waters in the Eastern Pacific. *Journal of Geophysical Research: Oceans*, 127, e2022JC018429. https://doi.org/10.1029/2022JC018429

Pardo P, Padín X, Gilcoto M, Farina-Busto L, Pérez F. 2011. Evolution of upwelling systems coupled to the long term variability in sea surface temperature and Ekman transport. Clim. Res. 48:231–46

Rykaczewski, R. R., J. P. Dunne, W. J. Sydeman, M. García-Reyes, B. A. Black, and S. J. Bograd (2015), Poleward displacement of coastal upwelling-favorable winds in the ocean's eastern boundary currents through the 21st century, *Geophys. Res. Lett.*, 42, 6424–6431, doi:10.1002/2015GL064694.

Reply

---

## Referee Report (RR1)

**Review of manuscript egusphere-2024-836** "Reviews and synthesis on increasing hypoxia in eastern boundary upwelling systems: zooplankton under metabolic stress"

The manuscript titled "Reviews and synthesis on increasing hypoxia in eastern boundary upwelling systems: zooplankton under metabolic stress" provides a comprehensive and timely synthesis of the effects of hypoxia on zooplankton, with a focus on Eastern Boundary Upwelling Systems (EBUS), a critical area affected by climate change. Zooplankton play a vital role in marine food webs, and understanding their physiological and ecological responses to hypoxia is crucial for predicting broader ecosystem impacts. The review integrates numerous studies and presents valuable insights into the effects of deoxygenation on marine organisms. However, while it addresses relevant and pressing topics, the manuscript suffers from unclear, overly complex, and repetitive writing at times. The scientific argumentation lacks depth in discussing the mechanistic links between hypoxia and ecosystem impacts, and some key ideas are introduced without sufficient clarification or support. After addressing the detailed comments regarding clarification of mechanisms, inclusion of quantitative data, improved structure, and a more thorough discussion of ecosystem consequences, this review will become a robust resource for researchers and policymakers. Once these revisions are made, the paper should be published, as it will make a significant contribution to the fields of marine ecology and climate change research.

**Major Points of Critique :**

1. **Vagueness in Hypoxia Mechanisms**: The paper frequently mentions that hypoxia affects zooplankton through physiological and behavioral responses but lacks detailed explanations on how exactly these mechanisms operate at a biochemical or ecological level. For example, the metabolic consequences of hypoxia and oxidative stress are only superficially discussed without clearly linking these processes to specific outcomes for zooplankton populations or ecosystems. Including this information would provide a very strong motivation for the review and provide and excellent resource for future researchers.
2. **Insufficient Discussion of Data and Citations**: Some sections mention important concepts (e.g., "variable tolerance to hypoxia") without referring to specific datasets or detailed empirical evidence. As before, providing concrete examples are needed to motivate the review and support broad claims. Additionally, some of the claims appear to lack references.
3. **Lack of Quantitative Insight**: The paper rarely provides quantitative information regarding the extent of hypoxia, or the physiological thresholds of different species, which could help readers understand the magnitude of the impact. These could be as simple as providing how the extent of the OMZ has changed globally in the last 20 years, or how it changes from season to season, and providing some species-specific thresholds for key species presented in the manuscript.

**Suggested Improvements:**

1. **Clarification of Hypoxia Mechanisms**: The manuscript should go deeper into the physiological responses of zooplankton to low oxygen. For instance, the authors could describe, how oxidative stress impacts key processes such as reproduction or energy metabolism. Afterwards, the authors could describe by how much reproduction

decreases as a function of the experienced oxygen concentration. An explanation of how different metabolic pathways (aerobic vs. anaerobic) are triggered in various conditions would also enhance clarity.

2. **Improving Structure and Reducing Redundancy**: The manuscript tends to repeat certain points about hypoxia and upwelling processes, especially in the introduction and review of adaptive responses. Condensing these points and organizing the material more logically would improve the flow. For example, after introducing the role of zooplankton in the food web, the physiological responses could be systematically addressed before transitioning to broader ecological consequences.

3. **Addition of Quantitative Examples**: Including more specific data points, such as oxygen levels in different EBUS regions and corresponding zooplankton adaptations, would greatly enhance the argument. Graphs or summary tables showing hypoxia thresholds for key species, and how these relate to oxygen availability in upwelling zones, would make the paper more informative.

4. **Expand the Ecological Consequences Section**: The section discussing ecological impacts is underdeveloped. The paper should explore the consequences of zooplankton stress for higher trophic levels in more detail, specifically discussing how changes in vertical migration or metabolic rates translate into shifts in predator-prey dynamics or nutrient cycling.

5. **Recommendation for Conceptual Framework**: I suggest the authors to develop a conceptual framework or model (perhaps visual) that links hypoxia, zooplankton physiology, behavior, and ecosystem impacts. This would provide structure and clarity, while helping unify the various points discussed.

6. **Highlight Methodological Advances**: I encourage the authors to discuss emerging technologies or methodologies (e.g., genomic studies, real-time oxygen tracking) that could enhance understanding of hypoxia in EBUS systems, making the review more forward-looking.

**Language and Grammatical Errors:**

1. **Vague Phrasing**: Terms like "hypoxia can challenge marine life" and "it is well known that zooplankton are affected" should be made more precise by specifying what kind of challenge (e.g., reduced reproductive success, mortality) and what specific zooplankton groups are affected.

2. **Redundancy**: Some sentences, such as the discussion of oxygen minimum zones and upwelling intensification, can be condensed to avoid redundancy.

3. **Unclear Sentence Structures**: Sentences such as "The ongoing combined processes, deoxygenation, increasing upwelling, and OMZ expansion will alter the oxygen conditions in upper layers (<50 m) in EBUS" are complex and hard to follow. Breaking them into simpler sentences would improve readability.

4. **Spelling and Punctuation**:
   o Many times the verb does not fit the subject (has instead of have).
   o The term "deoxygenation" is repeated redundantly in some sections. Using alternative phrasing like "oxygen loss" could improve readability.
   o A few commas and conjunctions are missing in sentences, making them unnecessarily complicated.

5. **Technical Terms Without Definition**: Terms such as "Pcrit" and "ROS" should be explained more thoroughly when first introduced to ensure clarity for all readers.

6. **Inconsistent Citations**: Some important statements lack citations, while others rely on the same sources repeatedly without diversifying evidence. Consider diversifying references, especially when dealing with recent studies.

**Detailed comments:**

L 17-18: Revise punctuation and phrasing. For instance: These effects, however, depend on specific adaptations of organisms that have evolved in habitats that are permanently or episodically subjected to low oxygen waters.

L 20: the oxidative stress is derived from the exposure to highly fluctuating oxygen conditions, rather than from the conditions themselves.

L 28-29: Stating that increases in mean global sea surface are a consequence of the warming of Earth's surface and ocean feels redundant. Please rephrase.

L 31-32: Use active voice as this is easier to understand. For instance: The warming of the upper layers of the ocean also drives a greater stratification of the water column, which reduces vertical mixing and affects ocean ventilation. A warmer ocean also lowers oxygen solubility, which further challenges marine life.

L 33-34: Do you mean by about 2%? Also, you specify "a decline in oxygen" and "oxygen loss" in the same sentence, which feels redundant.

L 37-38: The authors should explain why the decrease in oxygen is more critical. If OMZs are already present, them expanding might not necessarily be worse than a lowering of oxygen at a place where oxygen was previously at a high concentration.

L 39: I encourage the authors to be more quantitative here. For instance, they could specify the area and provide specific locations.

L 42: It is not clear what is meant by "intensifying the OMZ". Do you mean the extent gets larger, or that the oxygen concentration is lowered, or both?

L 43-44: Remove "the action of". Name in the citation is wrong according to the bibliography at the end (same error in L 57)

L 57: Should be "…takes place, which compresses the …"

Figure 1: It should be raises rather than rises, as the upwelling is performing the action.

Figure 2: I encourage the authors to describe the figures in the captions with more detail. As such, the authors could name both modes by name and describe the critical oxygen tension in this caption, including the colors used. In addition, I encourage the users to avoid using green and red in the same plot as this is the most common type of color vision deficiency.

L 109: Only short timescales rather than short-term timescales

L 113: Comma before which

Figure 3: Consistent labels Figure 3 instead of Fig. 3. Also avoid using red and green in the same plot, and a description of the shading is missing.

L 128-131: Here the authors describe a lot of mechanisms and the interactions with the metabolic rate. However, they are not specific on how everything changes and makes it difficult to understand. It might be easier to say how MR and Pcrit specifically change as oxygen increases/decreases.

L 133: Rephrase "does not allow taking advantage of …". Furthermore, the response itself cannot take advantage of favourable periods. Rather the organisms with this adaptive response can do so.

L. 133-135: Rephrase. The verb does not match the subject. It might also be better to rephrase it from the point of view of an organism. For instance: "Organisms with the second adaptive response are able to exploit …".

L. 137: Please rephrase. For instance: "Pcrit is constant within a given fish species."

L 141: Punctuation before "Furthermore" missing and it should be "each" rather than "along"

L. 142-143: Rephrase and use the active voice. The verb in the second half does not match.

L 162: Please rephrase and use the active voice.

L 164-166: This sentence stands in contrast with the previous one where the authors describe that M. norvergica is able to resist prolonged periods of low oxygen conditions.

L 170: Rephrase. "…behavioural adaptations have been observed…"

L 170-172: This sentence is convoluted and difficult to understand. Please rephrase.

Section 3: I encourage the authors to structure this section into several subsections, as this would increase readability and clarity.

L 174: Start with "Oxidative stress…" as this would make the sentence much clearer.

L. 176-179: The explanation should come at the beginning of the paragraph.
L 179: It is unclear what is meant by "stressful"

L 184-185: Please rephrase. Rather than saying whether this is surprising or not, I encourage the authors to directly tell the reader why this is important. For instance: "Since all signaling molecules are highly dependent on the oxygen availability for mitochondrial functioning, their regulation is likely impaired by the exposure to variable oxygen levels  near the OMZ".

L 191: O2 was introduced before already

L 218: POS was presented previously already.

L 225. Remove the first sentence or integrate in the second one, as it does not add any information to the manuscript.

L 225-226: Please specify less stressful than what and better than what or under which conditions.

L 234-235: Please rephrase. This sentence is difficult to understand.

L 239: I encourage the authors to explain what the timing of the development of antioxidant defences is important. Here it would be important to link the development of antioxidant defences to population dynamics and compare the life-stage compositions of key species and potential exposure to deoxygenated waters across the seasons.

L 251: Remove the sentence "The interplay between …" or incorporate into the previous sentence as (see Figure X).

L 251-254: Please rephrase. This sentence is rather long and has a lot of information that could be split into several sentences.

L 267: I would rephrase to something along the lines of "likely led to the adaptation" or "likely selected for the adaptation" rather than "required the evolution".

L 270-273: I encourage the authors to include some references in this sentence.

L 271: SDA is only used once.

L 274-276: Please rephrase. For instance: "However, not all zooplankton perform DVM where for instance copepods, which significantly contribute to the bulk of zooplankton biomass in the global ocean, remain within the upper 50 m of the water column."

L 279-280: Remove "Also, as mentioned above" and combine with the previous sentence. In addition, remove occasional, as extreme events by definitions are not frequent.

L 281: Rephrase to "has a strong seasonal signal" or "has a strong seasonal variation"

L 285: Verb does not match subject. Do you mean Autumn-Winter winds/conditions exhibit…?

L 285: Please rephrase. In my opinion, "depressed upwelling" is not necessarily the same as downwelling.

L 287-288: Please rephrase. For instance: "before the upwelling spin-up phase" or "during the transitions phase between downwelling and upwelling"

L 290: What is meant by projected conditions? I was under the impression that this paragraph was about upwelling vs. downwelling.

L 290: Capitalize to match the previous use of "Autumn-Winter" (L 285)

L 291: In my opinion it should be "zooplankton are"

L 292-294: Please rephrase: "This oxic condition promotes the existence of ROS production, whereas strong upwelling and the shoaling of the OMZ into the photic zone can expose non-migratory zooplankton to changing hypoxia-normoxia conditions, which result in POS+ROS."

L 295: Use "by further compressing the vertical extent of the oxygenated habitat"

Figure 6: Spell out OMZ in the caption.

L 304-306: This paragraph feels out of place here.

L 315: The fact that zooplankton tend to have short life spans should be introduced sooner in the introduction. This way, the reader would be able to follow the argumentation here better.

L 344-347: Please rephrase. This sentence has a lot of information and is difficult to understand.

---

## Author Response (AR2)

LETTER OF RESPONSE TO REVIEWERS

Dear Editor

Biogeosciences

We are here submitting a revised version of the MS "Reviews and synthesis on increasing hypoxia in eastern boundary upwelling systems: zooplankton under metabolic stress", by Frederick and co-authors. We thank you and the reviewer for the constructive criticism, we have considered all comments, corrections and suggestions from the reviewer, which are now included in the present version.

We appreciate the consideration of our work and thank the reviewer for the very detailed revision, which has considerably improved our work.

On behalf of all coauthors

Ruben Escribano

Review of manuscript egusphere-2024-836 "Reviews and synthesis on increasing hypoxia in eastern boundary upwelling systems: zooplankton under metabolic stress" The manuscript titled "Reviews and synthesis on increasing hypoxia in eastern boundary upwelling systems: zooplankton under metabolic stress" provides a comprehensive and timely synthesis of the effects of hypoxia on zooplankton, with a focus on Eastern Boundary Upwelling Systems (EBUS), a critical area affected by climate change. Zooplankton play a vital role in marine food webs, and understanding their physiological and ecological responses to hypoxia is crucial for predicting broader ecosystem impacts. The review integrates numerous studies and presents valuable insights into the effects of deoxygenation on marine organisms. However, while it addresses relevant and pressing topics, the manuscript suffers from unclear, overly complex, and repetitive writing at times. The scientific argumentation lacks depth in discussing the mechanistic links between hypoxia and ecosystem impacts, and some key ideas are introduced without sufficient clarification or support. After addressing the detailed comments regarding clarification of mechanisms, inclusion of quantitative data, improved structure, and a more thorough discussion of ecosystem consequences, this review will become a robust resource for researchers and policymakers.

 Once these revisions are made, the paper should be published, as it will make a significant contribution to the fields of marine ecology and climate change research.

**R. We thank the reviewer for the comments and suggestions. We have fully revised the text and deepened the analysis and discussion on mechanisms linking adaptive responses to hypoxia and consequences at population, community and ecosystem level.**

Major Points of Critique :

1. Vagueness in Hypoxia Mechanisms: The paper frequently mentions that hypoxia affects zooplankton through physiological and behavioral responses but lacks detailed explanations on how exactly these mechanisms operate at a biochemical or ecological level. For example, the metabolic consequences of hypoxia and oxidative stress are only superficially discussed without clearly linking these processes to specific outcomes for zooplankton populations or ecosystems. Including this information would provide a very strong motivation for the review and provide and excellent resource for future researchers.

**R. Thanks for the comment, we have further developed a mechanistic link between the physiological/biochemical responses/adaptions and the changes at population and whole ecosystem level. In this work we are stressing that the major effects of hypoxia on zooplankton are linked to metabolic responses which may depend on species adaptations or lack of adaptation to variable levels of oxygen, and the specific mechanisms result from a trade-off between the adoption of an oxygen-conforming or oxygen-regulating physiology and the energetic demands for other vital processes, such as reproduction, feeding or swimming to avoid predation. Regarding oxidative stress we here emphasize that exposure to severe hypoxia may lead to stressful conditions causing damage due to oxidative stress upon thereafter re-oxygenation. This can occur differently depending on the migration of some species for example, but regardless of that, ocean deoxygenation can certainly enhance this phenomenon with ecological consequences. We have now revised the text to make clear these messages. In the same context, we are adding a new Figure in which we describe in detail the mechanisms underlying hypoxia effects and the connections among different ecological levels.**

2. Insufficient Discussion of Data and Citations: Some sections mention important concepts (e.g., "variable tolerance to hypoxia") without referring to specific datasets or detailed empirical evidence. As before, providing concrete examples are needed to motivate the review and support broad claims. Additionally, some of the claims appear to lack references.

**R. We agree with the reviewer and now we are providing references and data for some mentions. New references have been added to the work.**

3. Lack of Quantitative Insight: The paper rarely provides quantitative information regarding the extent of hypoxia, or the physiological thresholds of different species, which could help readers understand the magnitude of the impact. These could be as simple as providing how the extent of the OMZ has changed globally in the last 20 years, or how it changes from season to season, and providing some species-specific thresholds for key species presented in the manuscript.

**R. We are now adding more quantitative data regarding hypoxia and responses to it. Variability of the OMZ over temporal scales are now included as well.**

Suggested Improvements:

1. Clarification of Hypoxia Mechanisms: The manuscript should go deeper into the physiological responses of zooplankton to low oxygen. For instance, the authors could describe, how oxidative stress impacts key processes such as reproduction or energy metabolism. Afterwards, the authors could describe by how much reproduction decreases as a function of the experienced oxygen concentration. An explanation of how different metabolic pathways (aerobic vs. anaerobic) are triggered in various conditions would also enhance clarity.

**R. See response above. We have now illustrated these physiological responses and how they impact energy usage by metabolism in a new Figure, which is also now discussed along the text.**

2. Improving Structure and Reducing Redundancy: The manuscript tends to repeat certain points about hypoxia and upwelling processes, especially in the introduction and review of adaptive responses. Condensing these points and organizing the material more logically would improve the flow. For example, after introducing the role of zooplankton in the food web, the physiological responses could be systematically addressed before transitioning to broader ecological consequences.

**R. Text has been fully revised, including these suggestions.**

3. Addition of Quantitative Examples: Including more specific data points, such as oxygen levels in different EBUS regions and corresponding zooplankton adaptations, would greatly enhance the argument. Graphs or summary tables showing hypoxia thresholds for key species, and how these relate to oxygen availability in upwelling zones, would make the paper more informative.

**R. We have revised the text and further looked for more specific and quantitative data. Some references have been added.**

4. Expand the Ecological Consequences Section: The section discussing ecological impacts is underdeveloped. The paper should explore the consequences of zooplankton stress for higher trophic levels in more detail, specifically discussing how changes in vertical migration or metabolic rates translate into shifts in predator-prey dynamics or nutrient cycling.

**R. A new figure illustrate potential impact at different ecological levels. The issue relating metabolic stress and further consequences in higher trophic levels in discussed along with a new Figure. However, how this can impact prey-predator interactions becomes an issue very difficult to address and we are just mentioning it but avoiding speculations about it.**

5. Recommendation for Conceptual Framework: I suggest the authors to develop a conceptual framework or model (perhaps visual) that links hypoxia, zooplankton physiology, behavior, and ecosystem impacts. This would provide structure and clarity, while helping unify the various points discussed.

**R. A new Figure has been added which represents a conceptual model on how metabolism, adaptive responses and energy demands from aerobic respirations can interact to ultimately affect at the ecosystem level.**

6. Highlight Methodological Advances: I encourage the authors to discuss emerging technologies or methodologies (e.g., genomic studies, real-time oxygen tracking) that could enhance understanding of hypoxia in EBUS systems, making the review more forward-looking.

**R. We are now providing some information on the subject in the Discussion section.**

Language and Grammatical Errors:

1. Vague Phrasing: Terms like "hypoxia can challenge marine life" and "it is well known that zooplankton are affected" should be made more precise by specifying what kind of challenge (e.g., reduced reproductive success, mortality) and what specific zooplankton groups are affected.

**R. Ok. Changed**

2. Redundancy: Some sentences, such as the discussion of oxygen minimum zones and upwelling intensification, can be condensed to avoid redundancy.

**R. The text has been revised and several paragraphs shortened or removed to avoid redundancy.**

3. Unclear Sentence Structures: Sentences such as "The ongoing combined processes, deoxygenation, increasing upwelling, and OMZ expansion will alter the oxygen conditions in upper layers (<50 m) in EBUS" are complex and hard to follow. Breaking them into simpler sentences would improve readability.

**R. Paragraph modified and corrected**

4. Spelling and Punctuation: o Many times the verb does not fit the subject (has instead of have). o The term "deoxygenation" is repeated redundantly in some sections. Using alternative phrasing like "oxygen loss" could improve readability. o A few commas and conjunctions are missing in sentences, making them unnecessarily complicated.

 **R. Text revised**

5. Technical Terms Without Definition: Terms such as "Pcrit" and "ROS" should be explained more thoroughly when first introduced to ensure clarity for all readers.

**R. Agreed. We have done so**

6. Inconsistent Citations: Some important statements lack citations, while others rely on the same sources repeatedly without diversifying evidence. Consider diversifying references, especially when dealing with recent studies.

**R. We have fully revised references and added cites**

Detailed comments: L 17-18: Revise punctuation and phrasing. For instance: These effects, however, depend on specific adaptations of organisms that have evolved in habitats that are permanently or episodically subjected to low oxygen waters.

**R. Ok. Corrected**

L 20: the oxidative stress is derived from the exposure to highly fluctuating oxygen conditions, rather than from the conditions themselves.

**R. Corrected. We modified the paragraph a separated these sentences.**

L 28-29: Stating that increases in mean global sea surface are a consequence of the warming of Earth's surface and ocean feels redundant. Please rephrase.

**R. Corrected. We deleted the second part**

L 31-32: Use active voice as this is easier to understand. For instance: The warming of the upper layers of the ocean also drives a greater stratification of the water column, which reduces vertical mixing and affects ocean ventilation. A warmer ocean also lowers oxygen solubility, which further challenges marine life.

**R. Agreed. Now modified**

L 33-34: Do you mean by about 2%? Also, you specify "a decline in oxygen" and "oxygen loss" in the same sentence, which feels redundant.

**R, Corrected**

L 37-38: The authors should explain why the decrease in oxygen is more critical. If OMZs are already present, them expanding might not necessarily be worse than a lowering of oxygen at a place where oxygen was previously at a high concentration.

**R. Agreed. We removed this paragraph to avoid redundancy**

L 39: I encourage the authors to be more quantitative here. For instance, they could specify the area and provide specific locations.

**R. Paragraph has been removed**

L 42: It is not clear what is meant by "intensifying the OMZ". Do you mean the extent gets larger, or that the oxygen concentration is lowered, or both?

**R. This paragraph was deleted to avoid redundancy**

L 43-44: Remove "the action of". Name in the citation is wrong according to the bibliography at the end (same error in L 57)

**R. OK. corrected**

L 57: Should be "...takes place, which compresses the ..." Figure 1: It should be raises rather than rises, as the upwelling is performing the action.

**R. This sentence was removed.**

Figure 2: I encourage the authors to describe the figures in the captions with more detail. As such, the authors could name both modes by name and describe the critical oxygen tension in this caption, including the colors used. In addition, I encourage the users to avoid using green and red in the same plot as this is the most common type of color vision deficiency.

**R. Agreed. The Figure was modified as suggested and the caption too to describe it in more detail.**

L 109: Only short timescales rather than short-term timescales L 113: Comma before which

**R. Corrected**

Figure 3: Consistent labels Figure 3 instead of Fig. 3. Also avoid using red and green in the same plot, and a description of the shading is missing.

**R. Corrected**

L 128-131: Here the authors describe a lot of mechanisms and the interactions with the metabolic rate. However, they are not specific on how everything changes and makes it difficult to understand. It might be easier to say how MR and Pcrit specifically change as oxygen increases/decreases.

**R. Agreed. This paragraph has been fully modified.**

L 133: Rephrase "does not allow taking advantage of …". Furthermore, the response itself cannot take advantage of favourable periods. Rather the organisms with this adaptive response can do so.

**R. Agreed. Modified as suggested**

L. 133-135: Rephrase. The verb does not match the subject. It might also be better to rephrase it from the point of view of an organism. For instance: "Organisms with the second adaptive response are able to exploit ….".

**R. This sentence was removed.**

L. 137: Please rephrase. For instance: "Pcrit is constant within a given fish species."

**R. Agree. Done**

L 141: Punctuation before "Furthermore" missing and it should be "each" rather than "along"

**R. Now corrected**

L. 142-143: Rephrase and use the active voice. The verb in the second half does not match.

**R. Ok. Corrected**

L 162: Please rephrase and use the active voice.

**R. Ok. the sentence has been changed**

L 164-166: This sentence stands in contrast with the previous one where the authors describe that M. norvergica is able to resist prolonged periods of low oxygen conditions.

**R: Agree. We have modified the paragraph**

L 170: Rephrase. "…behavioural adaptations have been observed…"

**R. Done**

L 170-172: This sentence is convoluted and difficult to understand. Please rephrase. Section 3: I encourage the authors to structure this section into several subsections, as this would increase readability and clarity.

**R. Paragraph modified**

 L 174: Start with "Oxidative stress…" as this would make the sentence much clearer.

**R. Agree. Done**

L. 176-179: The explanation should come at the beginning of the paragraph. L 179: It is unclear what is meant by "stressful"

**R. We have fully rephrased these paragraphs**

L 184-185: Please rephrase. Rather than saying whether this is surprising or not, I encourage the authors to directly tell the reader why this is important. For instance: "Since all signaling molecules are highly dependent on the oxygen availability for mitochondrial functioning, their regulation is likely impaired by the exposure to variable oxygen levels  near the OMZ".

**R. Agree. Done**

L 191: $O_2$ was introduced before already L 218: POS was presented previously already. L 225. Remove the first sentence or integrate in the second one, as it does not add any information to the manuscript.

**R.  Weh have checked all the text to avoid repetition.**

L 225-226: Please specify less stressful than what and better than what or under which conditions.

**R. Ok. Corrected.**

L 234-235: Please rephrase. This sentence is difficult to understand.

**R. Paragraph modified**

L 239: I encourage the authors to explain what the timing of the development of antioxidant defences is important. Here it would be important to link the development of antioxidant defences to population dynamics and compare the life-stage compositions of key species and potential exposure to deoxygenated waters across the seasons.

**R.  We used the word "timing" mistakenly, because we meant "time". Now changed.**

L 251: Remove the sentence "The interplay between …" or incorporate into the previous sentence as (see Figure X).

**R. Sentence removed**

L 251-254: Please rephrase. This sentence is rather long and has a lot of information that could be split into several sentences.

**R. Sentence removed**

L 267: I would rephrase to something along the lines of "likely led to the adaptation" or "likely selected for the adaptation" rather than "required the evolution".

**R. Agreed. Changed**

L 270-273: I encourage the authors to include some references in this sentence. L 271: SDA is only used once.

**R. This sentence has been removed (no reference required)**

L 274-276: Please rephrase. For instance: "However, not all zooplankton perform DVM where for instance copepods, which significantly contribute to the bulk of zooplankton biomass in the global ocean, remain within the upper 50 m of the water column."

**R. Agree. Done**

L 279-280: Remove "Also, as mentioned above" and combine with the previous sentence. In addition, remove occasional, as extreme events by definitions are not frequent.

**R. Paragraph has been modified**

L 281: Rephrase to "has a strong seasonal signal" or "has a strong seasonal variation" L 285: Verb does not match subject. Do you mean Autumn-Winter winds/conditions exhibit…?

**R. Agreed. Changed**

L 285: Please rephrase. In my opinion, "depressed upwelling" is not necessarily the same as downwelling.

**R. Agreed. We deleted depressed upwelling**

L 287-288: Please rephrase. For instance: "before the upwelling spin-up phase" or "during the transitions phase between downwelling and upwelling"

**R. Ok. Rephrased**

L 290: What is meant by projected conditions? I was under the impression that this paragraph was about upwelling vs. downwelling.

**R. We deleted this sentence**

L 290: Capitalize to match the previous use of "Autumn-Winter" (L 285) L 291: In my opinion it should be "zooplankton are"

**R. Done**

L 292-294: Please rephrase: "This oxic condition promotes the existence of ROS production, whereas strong upwelling and the shoaling of the OMZ into the photic zone can expose nonmigratory zooplankton to changing hypoxia-normoxia conditions, which result in POS+ROS."

**R. Done**

L 295: Use "by further compressing the vertical extent of the oxygenated habitat" Figure 6: Spell out OMZ in the caption.

**R. Done**

L 304-306: This paragraph feels out of place here. L 315: The fact that zooplankton tend to have short life spans should be introduced sooner in the introduction. This way, the reader would be able to follow the argumentation here better.

**R. Agreed. Sentence removed and idea presented before**

L 344-347: Please rephrase. This sentence has a lot of information and is difficult to understand.

**R. Agreed. The paragraph has been modified and shortened.**

---

## Author Response (AR3)

LETTER OF RESPONSE

07 January 2025

Dear Editor

Biogeosciences

We here submit the corrected version of the MS "Reviews and synthesis on increasing hypoxia in eastern boundary upwelling systems: zooplankton under metabolic stress", by Frederick and co-authors.

We have added the following items to the manuscript:

AUTHOR CONTRIBUTIONS

ACKNOWLEDGEMENTS (including our thanks to reviewers)

FUNDING SUPPORT

We thank you and appreciate the consideration of our work

 On behalf of all coauthors

Sincerely,

Ruben Escribano